# A high-resolution Global Aviation emissions Inventory based on ADS-B (GAIA) for 2019 – 2021

Roger Teoh[1], Zebediah Engberg[2], Marc Shapiro[2], Lynnette Dray[3] and Marc E.J. Stettler[1]

[1]Department of Civil and Environmental Engineering, Imperial College London, London, SW7 2AZ, United Kingdom
[2]Breakthrough Energy, 4110 Carillon Point, Kirkland, WA 98033, United States
[3]Air Transportation Systems Laboratory, School of Environment, Energy and Resources, University College London, London, WC1E 6BT, United Kingdom

*Correspondence to*: Marc E.J. Stettler ([m.stettler@imperial.ac.uk](mailto:m.stettler@imperial.ac.uk))

**Abstract.** Aviation emissions that are dispersed into the Earth's atmosphere affect the climate and air pollution, with significant spatiotemporal variation owing to heterogeneous aircraft activity. In this paper, we use historical flight trajectories derived from Automatic Dependent Surveillance–Broadcast (ADS-B) telemetry and reanalysis weather data for 2019–2021 to develop the Global Aviation emissions inventory based on ADS-B (GAIA). In 2019, 40.2 million flights collectively travelled 61 billion kilometres using 283 Tg of fuel, leading to $CO_2$, $NO_X$, non-volatile particulate matter (nvPM) mass and number emissions of 893 Tg, 4.49 Tg, 21.4 Gg, and $2.8×10^{26}$, respectively. Global responses to COVID-19 led to reductions in the annual flight distance flown, $CO_2$, and $NO_X$ emissions in 2020 (-43%, -48% and -50%, respectively relative to 2019) and 2021 (-31%, -41% and -43%, respectively) with significant regional variability. Short-haul flights with duration < 3 h accounted for 83% of all flights, yet only for 35% of the 2019 $CO_2$ emissions, while long-haul flights with duration > 6 h (5% of all flights) were responsible for 43% of $CO_2$ and 49% of $NO_X$ emissions. Globally, actual flight trajectories flown are, on average, ~5% greater than the great-circle path between the origin-destination airport but this varies by region and flight distance. An evaluation of 8,705 unique flights between London and Singapore showed large variabilities in the flight trajectory profile, fuel consumption and emission indices. GAIA captures the spatiotemporal distribution of aviation activity and emissions and is provided for use in future studies to evaluate the negative externalities arising from global aviation.

## 1 Introduction

Aviation enables the rapid movement of people and goods and contributes to approximately \$3.5 trillion or 4.1% of the global gross domestic product in 2018 (ATAG, 2020). Between 1970 and 2019, global revenue passenger kilometres (RPK) grew by 6.2% per annum (Airlines for America, 2022). COVID-19 related travel restrictions reduced global RPK by 66% in 2020, however global aviation activity is forecast to return to pre-pandemic levels by 2023 and continue to grow at 4% per annum until 2040 (Boeing, 2021; Airbus, 2021; Airlines for America, 2022).

While aviation generates significant economic and social benefits, it is also responsible for negative externalities in the form of contributions to climate change, noise, and air pollution. Global aviation $CO_2$ emissions in 2018 (~1034 Tg) accounted for approximately 2.4% of anthropogenic greenhouse gas emissions (Lee et al., 2021). However, in the absence of a large-scale shift to alternative aviation fuels, this figure is expected to increase over time due to the expected growth in air travel and decarbonisation of other industries. In addition, aviation also contributes to non-$CO_2$ climate impacts resulting from nitrogen oxide ($NO_X$), water vapour, sulphate and soot particle emissions (Lee et al., 2021). $NO_X$, which includes both nitric oxide (NO) and nitrogen dioxide ($NO_2$) gases, emitted in the stratosphere facilitates the production of: (i) ozone, which causes a warming effect; and (ii) hydroxyl radicals, which partly offsets this warming effect through the destruction of methane (Fuglestvedt et al., 1999; Myhre et al., 2011). Sulphate particles reflect incoming solar radiation (Penner et al., 1999), while soot particles, which consist of a mixture of black carbon, metallic compounds and organic particles (Petzold et al., 2013), can absorb incoming short-wave radiation and trap outgoing longwave radiation (Bond et al., 2013; Penner et al., 1999). Water vapour and soot particles can also lead to the formation of condensation trails (contrails) when conditions in the exhaust plume fulfil the Schmidt-Appleman criterion (Schumann, 1996). Under ice-supersaturated conditions, contrails can persist, spread and transition into contrail cirrus (Haywood et al., 2009). These non-$CO_2$ components, apart from sulphates, are thought to have a net warming effect with an effective radiative forcing (ERF) of 67 [21, 111] mW m$^{-2}$ (5–95% confidence interval), which corresponds to two-thirds of aviation's net ERF in 2018 (101 [55, 145] mW m$^{-2}$) (Lee et al., 2021). Emissions of $NO_X$ and particulate matter also lead to air quality degradation and adverse health outcomes (Stettler et al., 2011; Yim et al., 2013), and global aviation emissions at cruise altitudes are estimated to be responsible for ~8000 premature mortalities per year amounting to ~1% of air quality-related premature mortalities globally (Barrett et al., 2010).

As with other sectors, aviation emissions can be quantified using "top-down" or "bottom-up" approaches. Top-down estimates are based on the global jet fuel consumption provided by the International Energy Agency (IEA) (IEA, 2020), including fuel used in helicopters, military and general aviation, ground-based operations (i.e., auxiliary power units and airport ground vehicles) and engine testing (Olsen et al., 2013a; Stettler et al., 2011). Conversely, bottom-up approaches estimate the fuel consumption and emissions from individual flights based on some aircraft activity data, such as flight schedules (Baughcum et al., 1996; Sutkus et al., 2001; Owen et al., 2010; Eyers et al., 2005; Roof et al., 2007; Wilkerson et al., 2010; Simone et al., 2013; Wasiuk et al., 2016; Quadros et al., 2022). These bottom-up emissions inventories are commonly used with climate models to quantify the non-$CO_2$ impacts resulting from $NO_X$, soot, and contrails (Olsen et al., 2013b; Chen and Gettelman, 2013; Burkhardt and Kärcher, 2011; Bock and Burkhardt, 2019), as radiative forcing is highly sensitive to the spatiotemporal distribution of these pollutants, contemporaneous ambient meteorological conditions, and background atmospheric composition (Skowron et al., 2013; Bond et al., 2013; Teoh et al., 2022; Lee et al., 2021; Schumann et al., 2012). However, these bottom-up emission inventories are known to underestimate the total fuel consumption by 10–20% relative to top-down estimates (Olsen et al., 2013a; Quadros et al., 2022) because: (i) aircraft movements from military operations, non-scheduled traffic and general aviation are not completely captured (Owen et al., 2010; Wilkerson et al., 2010; Simone et al., 2013; Wasiuk

et al., 2016); (ii) missing flight trajectories and segments are filled with a great circle path in regions where radar coverage or tracking data is not available (Eyers et al., 2005; Wilkerson et al., 2010); (iii) for some datasets, all flights are flown with an idealised great circle trajectory at optimal altitudes and speeds between the origin-destination airport and the total distance is scaled up to account for routing inefficiencies (Simone et al., 2013; Wasiuk et al., 2016); (iv) fuel consumption in the landing and take-off cycle are assumed using standardised time-in-modes that might not be representative of real-world operations (Patterson et al., 2009); and (v) the use of monthly- or annually-averaged weather data (Simone et al., 2013), or International Standard Atmosphere (ISA) conditions (Kim et al., 2007) to calculate the fuel consumption and emissions.

Recently, airspace surveillance systems have been transitioning from radar tracking to the Automatic Dependent Surveillance-Broadcast (ADS-B) standard (Spire Aviation, n.d.; EUROCONTROL, 2021). Aircraft flying above 18,000 feet (~5.5 km) must be equipped with ADS-B transponders, where their GPS positions are automatically broadcasted twice per second and collected by a network of terrestrial and satellite-based receivers (ICAO, 2021a; Spire Aviation, n.d.). Therefore, ADS-B telemetry enables flights to be tracked at high spatiotemporal resolutions and over remote oceans, deserts, and mountain ranges with poor radar coverage (ICAO, 2021a; Spire Aviation, n.d.; EUROCONTROL, 2021), thereby addressing some limitations of existing emissions inventories (Eyers et al., 2005; Wilkerson et al., 2010; Simone et al., 2013; Wasiuk et al., 2016). Indeed, recent studies have demonstrated the feasibility of using ADS-B data to model aviation emissions at a flight- (Filippone et al., 2022, 2021; Wang et al., 2020), regional- (Klenner et al., 2022; Sun and Dedoussi, 2021; Filippone and Parkes, 2021; Zhang et al., 2022), and global-level (Quadros et al., 2022; Liu et al., 2020). While an ADS-B-based global aviation emissions inventory for 2017–2020 has been developed by Quadros et al. (2022), emissions were computed using monthly-averaged flight trajectories and wind vectors. As flight paths generally have a large day-to-day variability in real-world operations (Klenner et al., 2022), monthly aggregation might introduce inaccuracies due to mismatches between trajectories and meteorology and the spatiotemporal distribution of emissions.

In this paper, we simulate the fuel consumption and emissions from individual flights using reanalysis weather data and historical flight trajectories derived from an air traffic dataset containing global ADS-B telemetry from January-2019 to December-2021. These results are then used to: (i) produce a global aviation emissions inventory for 2019–2021; (ii) evaluate the impacts of COVID-19 to aviation emissions; and (iii) quantify the routing inefficiencies and distribution of aviation emissions with respect to the flight mission profile. The main aim of this research is to provide a global aviation emissions inventory that is spatiotemporally accurate and publicly available, so it can be used as inputs to various health and climate models to estimate the impacts that arise from global aviation activity. Further information not included in the main text are presented in the Supporting Information (SI) as referenced.

## 2 Materials and methods

We combine multiple datasets to produce the global aviation emissions inventory for 2019–2021 that is named as the Global Aviation Emissions Inventory based on ADS-B (GAIA). These include: (i) a global ADS-B aircraft activity dataset from a commercial company (Spire Aviation), hereby known as the ADS-B dataset; (ii) a fleet database from a commercial company (Cirium) providing the specific aircraft variant and engine model for all registered aircraft globally (Cirium, 2022); (iii) the Base of Aircraft Data Family 4.2 (BADA4) and 3.15 (BADA3) aircraft performance models (EUROCONTROL, 2019, 2016); (iv) the International Civil Aviation Organization (ICAO) Aircraft Engine Emissions Databank (EDB) that contains the emission indices from aircraft gas turbine engines with rated thrust above 26.7 kN (EASA, 2021); and (v) historical weather data from the European Centre for Medium-Range Weather Forecast (ECMWF) ERA5 high-resolution realisation (HRES) (ECMWF, 2021; Hersbach et al., 2020).

### 2.1 ADS-B dataset

Global aircraft ADS-B telemetry data for 2019–2021 were collected by combination of terrestrial receivers and commercial satellite constellation (Spire Aviation), with increasing global coverage over time (Fig. S1 in the SI). For each flight, the unique aircraft identifier (ICAO 24-bit aircraft address and call sign) and position (longitude, latitude, and altitude) are provided at a temporal resolution of 300 s. Spire Aviation augmented the dataset with third-party data sources and flight schedules, providing additional information such as the aircraft tail number, ICAO aircraft type designator, and the ICAO airport codes for the origin and destination airports.

The raw ADS-B telemetry may contain limitations where multiple unique flights can share the same identifier (Fig. S3) and/or have missing flight segments (Fig. S5 and S6b). We developed a data cleaning algorithm detailed in the SI §S1.2. In summary, the algorithm: (i) identifies and groups waypoints that belong to distinctive flights, ensuring that each unique flight contains waypoints with the same aircraft and flight properties (i.e., identifier, aircraft type, tail number, and origin-destination airport pair); (ii) fills any missing flight segments whenever possible; and (iii) verifies that the constructed flight trajectories have a realistic flight time, segment length, altitude profile, and ground speed. For each flight, a great-circle interpolation is performed between the recorded waypoints so that the resulting time differences between waypoints is between 40 and 60 s. The interpolation algorithm also incorporates step climbs/descents at cruise altitude to ensure that the interpolated trajectories conform to real-world airspace design and air traffic management constraints (Dalmau and Prats, 2017) (Fig. S5 and S6a). Flights with incomplete trajectories, i.e., the first (final) waypoint does not start (end) at the origin (destination) airport, are extrapolated using a great-circle path to the recorded airport whenever possible, or the nearest airport if data is not available (Fig. S6b). Flights with missing, anonymised, and unidentifiable aircraft types (i.e., rotorcraft and sensitive military flights), and/or unrealistic flight trajectories are removed from the database (~5% of all flights, Fig. S7e).

The processed ADS-B dataset contains 103.7 million unique flight trajectories between 2019 and 2021. 75.2% of these flights are carried out by jet aircraft, 9.4% by turboprops, and 15.4% by piston aircraft. Origin and destination airport metadata is available for 79% of all flights (92% of flights performed by jet aircraft), and 67.4% of all flights (77.6% of jet flights) have full trajectory coverage, i.e., waypoints starting from origin and ending at destination (Fig. S7). We assess the completeness of the ADS-B dataset by comparing the total number and flight distance flown with global statistics from ICAO, the International Air Transport Association (IATA), and Airlines for America (ICAO, 2020; ICAO, 2021b; IATA, 2022; Airlines for America, 2022) (SI §S1.3). As the statistics from ICAO, IATA and Airlines for America only captures air traffic activity from scheduled flights, we exclude general aviation activity in these comparisons by omitting flights that are flown by piston aircraft. The total number of flights in the ADS-B dataset differs by -4.7% (2019), +14% (2020), and +17% (2021), respectively, relative to the statistics from ICAO and IATA (Table S1); while the annual flight distance flown in the ADS-B dataset are 8% (2019), 23% (2020), and 24% (2021) larger than the estimates from Airlines for America (Table S2). These discrepancies are likely due to: (i) an increasing global coverage area of ADS-B receiver networks over time enabling more flights to be captured in the ADS-B dataset (Fig. S1); (ii) an increase in the proportion of non-scheduled flights, i.e., charter flights and private aviation, from 4.1% in 2019 to 7.5% in 2020 (Sobieralski and Mumbower, 2022; ICAO, 2021b); and (iii) a higher occurrence of rejected flights (i.e., trajectories with less than three waypoints, unrealistic segment lengths, flight times and/or ground speeds) in 2019 (~6.6%) relative to 2020 (~3.3%) and 2021 (~4.5%) (Fig. S7e). A comparison with data from three major airports suggest that the 2019–2021 air traffic movements in the ADS-B dataset is 1.3%, 7.0%, and 1.3% lower than the official statistics from London Heathrow, New York John F. Kennedy, and Singapore Changi airport, respectively (Fig. S8), and this discrepancy can most likely be attributed to our data cleaning algorithm which rejected flights with erroneous trajectories that cannot be verified (SI §S1.2).

## 2.2 Aircraft performance model

Aircraft mass and fuel consumption are estimated using BADA4 and BADA3 (EUROCONTROL, 2019, 2016). BADA4 provides aircraft characteristics and performance data for 105 aircraft-engine combinations, covering 55.1% of all flights and 78.3% of the total flight distance in the ADS-B dataset, and is used whenever possible because of its performance improvements relative to BADA 3 (Nuic et al., 2010). BADA 3 is used for the remaining flights, including flights that are flown by turboprop and piston aircraft. For each flight, we use the aircraft tail number registrations and year to obtain the specific aircraft variant and engine model if it is in the fleet database (Cirium, 2022). The fleet database covers 59% of all flights in the ADS-B dataset or 79% of flights carried out by jet aircraft, and a breakdown of the engine market share for commonly used aircraft types are provided in Table S3. For aircraft not covered by the fleet database, we assign the default aircraft-engine combination provided by BADA 3 with modifications applied to the Airbus A320, Boeing B787 to select the engine option with the highest market share (Table S4).

The ambient temperature and horizontal wind components are required to calculate the true airspeed and Mach number at each waypoint, c.f. Eq. (S1) and (S2) in the SI §S1.1, and we obtain the local meteorology by performing a quadrilinear interpolation against historical weather data from the ERA5 HRES reanalysis at a $0.25° \times 0.25°$ horizontal resolution over 37 pressure levels and at 1 h time resolution (ECMWF, 2021; Hersbach et al., 2020). The aircraft mass and fuel mass flow rate ($\dot{m}_f$) at each waypoint are then estimated iteratively until convergence (Wasiuk et al., 2015). In the initialisation run, the aircraft mass at the first waypoint ($M_0$) is set to the aircraft-specific maximum take-off weight (MTOW), the BADA total energy model (TEM) is used to estimate the $\dot{m}_f$ (EUROCONTROL, 2019, 2016), and the aircraft mass at subsequent waypoints decrease in accordance with fuel consumption. For subsequent iterations, $M_0$ is estimated as a function of the mission profile and passenger load factor,

$$M_0 = \text{OEW} + (\text{LF} \times \text{MPL}) + (M_f^{\text{mission}} + M_f^{\text{reserve}}), \text{ and} \tag{1}$$

$$M_0 = \min(M_0, MTOW),$$

where OEW is the aircraft operating empty weight, LF is the assumed passenger load factor, MPL is the maximum payload, and $M_f^{\text{mission}}$ is the total fuel consumption required to fly the mission profile that is estimated from the previous iteration. $M_f^{\text{reserve}}$ is the reserve fuel requirements (Wasiuk et al., 2015) that is approximated by taking the maximum of either the fuel required to fly the aircraft for an additional 90 minutes at the top of descent or 15% of $M_f^{\text{mission}}$. BADA provides the OEW, MPL and MTOW for specific aircraft types, and the upper limit of $M_0$ is constrained to MTOW.

The historical passenger LF is used to account for the effects of COVID-19 (Fig. S10): for all flights with airport metadata, the regional passenger LF is assigned based on the origin airport; while the global mean passenger LF is assumed when airport metadata is not available. Due to data limitations, our approach does not account for the LF variability between different airlines, aircraft size, and mission profile (i.e., short/long haul flights and passenger/freight services). An earlier study found that the simulated fuel consumption has a low sensitivity to LF assumptions, where varying their assumed annual mean LF (62.8%) by ±7.7% resulted in a ±1.1% change in global annual fuel consumption (Quadros et al., 2022).

### 2.3 Emissions

Engine-specific data are provided by the ICAO EDB to estimate the engine thrust settings, thermodynamic quantities at different stages of the engine, and emission indices (EI) of different pollutants (EASA, 2021). Table 1 summarises the methodologies that are used to estimate the $CO_2$, $NO_X$, carbon monoxide (CO), hydrocarbons (HC), organic carbon (OC), water vapour ($H_2O$), sulphur oxides ($SO_2$), sulphate particles ($S^{VI}$), and non-volatile particulate matter (nvPM) mass and number emissions. As the usage of sustainable aviation fuel remains low ($< 0.1\%$ of the annual fuel consumption in 2018) (Le Feuvre, 2019), we assume that all flights are powered by fossil kerosene fuel (Jet A-1) with a fuel sulphur content of 600 parts per million. $CO_2$, $H_2O$, OC, $SO_2$, and $S^{VI}$ emissions depend on the fuel properties and are estimated with a constant EI.

**Table 1: Summary of the methods/ emissions indices that are selected to calculate the $CO_2$ and non-$CO_2$ pollutants, where they are: (i) listed in order of priority; and (ii) quantified as a percentage of the total number of flights and percentage of the total flight distance flown in the processed ADS-B dataset.**

| Pollutant | Methods | Emissions index/Remarks | % of all flights | % of total flight distance | Ref. |
|---|---|---|---|---|---|
| $CO_2$ | | 3.159 kg kg$^{-1}$ | | | [1] |
| $H_2O$ | | 1.237 kg kg$^{-1}$ | | | [1] |
| OC | Constant | 20 mg kg$^{-1}$ | 100% | 100% | [2] |
| $SO_2$ | | 1.2 g kg$^{-1}$, 98% conversion efficiency of $SO_X$ to $SO_2$ | | | [3] |
| $S^{VI}$ | | 0.024 g kg$^{-1}$, 2% conversion efficiency of $SO_X$ to $S^{VI}$ | | | [1], [2] |
| NO$_X$ | 1. FFM2 | For engines where gaseous emissions are in the EDB. | 72.7% | 93.6% | [4] |
| | 2. Constant | 15.14 g kg$^{-1}$ otherwise | 27.3% | 6.4% | [3] |
| CO | 1. FFM2 | For engines where gaseous emissions are in the EDB. | 72.7% | 93.6% | [4] |
| | 2. Constant | 3.61 g kg$^{-1}$ otherwise | 27.3% | 6.4% | [3] |
| HC | 1. FFM2 | For engines where gaseous emissions are in the EDB. | 72.7% | 93.6% | [4] |
| | 2. Constant | 0.520 g kg$^{-1}$ otherwise | 27.3% | 6.4% | [3] |
| nvPM mass | 1. $T_4/T_2$ method | For engines where nvPM emissions are in the EDB. | 63.3% | 82.4% | [5] |
| | 2. FOX & ImFOX | For engines that are covered in the EDB. | 9.4% | 11.2% | [6], [7] |
| | 3. Constant | 0.088 g kg$^{-1}$ otherwise | 27.3% | 6.4% | [6] |
| nvPM number | 1. $T_4/T_2$ method | For engines where nvPM emissions is in the EDB. | 63.3% | 82.4% | [5] |
| | 2. FA model | For engines that are covered in the EDB, assumes the emissions profile of singular annular combustors. | 9.4% | 11.2% | [8] |
| | 3. Constant | $10^{15}$ kg$^{-1}$ otherwise | 27.3% | 6.4% | [8], [9] |

[1] Wilkerson et al. (2010); [2] Stettler et al. (2011); [3] Lee et al. (2021); [4] DuBois & Paynter (2006); [5] Teoh et al. (2022); [6] Stettler et al. (2013); [7] Abrahamson et al. (2016); [8] Teoh et al. (2020); [9] Schumann et al. (2015)

The EI for NO$_X$, CO and HC varies with engine operating conditions and combustor type and are estimated using: (i) the Fuel Flow Method 2 (FFM2) (DuBois and Paynter, 2006) when gaseous EI for the specific engine type is provided by the ICAO EDB (EASA, 2021) (available for 557 unique engine types, covering 72.7% of all flights and 93.6% of the flight distance in the ADS-B dataset); or (ii) a constant EI that is representative of the historical fleet-average values (Wilkerson et al., 2010) when engine-specific EI data is not available (27.3% of all flights, of which 91% of these are flights flown by turboprop and piston aircraft). The NO$_X$ EI is sensitive to the ambient humidity and a humid atmosphere can suppress engine NO$_X$ production (DuBois and Paynter, 2006). While the FFM2 assumes a fixed relative humidity of 60% for all waypoints, we use humidity data from the ERA5 HRES at each waypoint. We also highlight that the engine-specific NO$_X$ EI in the ICAO EDB is reported as an $NO_2$ mass equivalent (ICAO, 2017). For future studies that require cruise NO$_X$ emissions to be broken down into individual species, references can be made to previous in-situ measurements which assumes the engine exit $NO_2$/NO$_X$ and NO/NO$_X$ molar mixing ratio to have a global mean of 0.07 and 0.93 respectively (Schulte et al., 1997), and the nitrous acid (HONO) EI to be 0.31 g per kg-$NO_2$ (Jurkat et al., 2011). For the landing and take-off cycle (LTO), existing studies have estimated that the $NO_2$/NO$_X$ molar mixing ratio varies significantly based on engine type and thrust settings, and ranges

between: (i) 0.05–0.10 during climb and take-off; (ii) 0.12–0.20 during the descent phase; and (iii) 0.75–0.98 during the taxi phase (Timko et al., 2010; Wood et al., 2008; Wey et al., 2006; Stettler et al., 2011).

The emissions profile for the nvPM $EI_m$ and $EI_n$ are unique for different engine combustor type and power settings, and can vary by up to five orders of magnitude (EASA, 2021; Teoh et al., 2022). Since December-2020, the ICAO EDB started reporting the nvPM $EI_m$ and $EI_n$ for all in-production and new turbofan engines with rated thrust above 26.7 kN, covering 178

unique engine types (ICAO, 2022; EASA, 2021).We use the nvPM EI's that are corrected for dilution, thermophoretic and particle line losses (EASA, 2020). For aircraft-engine types with nvPM measurements (63.3% of all flights and 82.4% of the total flight distance), the nvPM $EI_m$ and $EI_n$ are estimated by linear interpolation relative to the ratio of turbine inlet to compressor inlet temperatures ($T_4/T_2$), a non-dimensional measure of engine thrust settings that captures the differences in engine operating conditions at ground and cruise (Cumpsty and Heyes, 2015). The $T_4/T_2$ methodology, originally developed

by Teoh et al. (2022), is improved and evaluated with ground and cruise nvPM $EI_n$ measurements from the ECLIF II/ND-MAX experimental campaign (Schripp et al., 2022; Bräuer et al., 2021; Voigt et al., 2021) (SI §S4.1). For older aircraft-engine types where nvPM is not reported in the ICAO EDB (9.4% of flights and 11.2% of the flight distance), we estimate the nvPM according to Teoh et al. (2020): the $EI_m$ is estimated by using an average of the Formation and Oxidation (FOX) (Stettler et al., 2013) and Improved FOX methods (Abrahamson et al., 2016), assuming the emissions profile of singular annular

combustors; and the fractal aggregates model is used to convert the estimated $EI_m$ to $EI_n$ (Teoh et al., 2019, 2020) (SI §S4.2). For the remaining flights without engine-specific data, the nvPM $EI_m$ and $EI_n$ are set to constant values of 0.088 g kg$^{-1}$ and $10^{15}$ kg$^{-1}$, respectively, which are nominal fleet-average values reported in earlier studies (Stettler et al., 2013; Schumann et al., 2015; Teoh et al., 2020).

**2.4 Global aviation emissions inventory**

GAIA is processed in three formats: (i) flight-waypoint outputs containing the fuel consumption and emissions at each waypoint; (ii) flight-summary outputs, which contain the metadata (flight identifier, mission profile, and aircraft-engine characteristics) and aggregate the fuel consumption and emissions for each flight; and (iii) gridded outputs that aggregate the flight-waypoint outputs to a 4D grid with 0.5° x 0.5° horizontal resolution x 1000 ft. altitude and at a 1 h temporal resolution. These different outputs will allow GAIA to be used as inputs to different climate models, such as Lagrangian-based models

that use flight-waypoint data (Schumann, 2012; Schumann et al., 2012; Caiazzo et al., 2017; Fritz et al., 2020) and general circulation models that work with gridded inputs (Olsen et al., 2013b; Chen and Gettelman, 2013; Burkhardt and Kärcher, 2011; Bock and Burkhardt, 2019). Here, the flight-waypoint outputs are used to compute statistics on the routing inefficiencies and distribution of aviation emissions by mission profile, while rectangular spatial bounding boxes are applied to the gridded outputs to estimate the aviation emissions from different world regions (Fig. 1 and Table 2). We note that GAIA only captures

the fuel consumption and emissions after the flight is airborne, and hence, do not account for ground emissions during taxi, take-off, and from the use of Auxiliary Power Units (APU).

**Table 2: Spatial bounding boxes used to estimate the regional aviation activity and emissions.**

| Region | Bounding box | | Surface area ($\times 10^{13}$ m²) | Global surface area | Source |
|---|---|---|---|---|---|
| USA | 126° W – 66° W | × 23° N – 50° N | 1.6005 | 3.1% | [1] |
| Europe | 12° W – 20° E | × 35° N – 60° N | 0.6662 | 1.3% | [1] |
| East Asia | 103° E – 150° E | × 15° N – 48° N | 1.6170 | 3.2% | [1] |
| Southeast Asia | 87.5° E – 130° E | × 10° S – 20° N | 1.5533 | 3.1% | [3] |
| Latin America | 85° W – 35° W | × 60° S – 15° N | 3.9774 | 7.8% | [3] |
| Africa & Middle East | 20° W – 50° E | × 35° S – 40° N | 6.0334 | 12% | [3] |
| China | 73.5° E – 135° E | × 18° N – 53.5° N | 2.1628 | 4.2% | [2] |
| India | 68° E – 97.5° E | × 8° N – 35.5° N | 0.9244 | 1.8% | [2] |
| North Atlantic | 70° W – 5° W | × 40° N – 63° N | 1.1493 | 2.3% | [1] |
| North Pacific | 140° E – 120° W | × 35° N – 65° N | 2.3577 | 4.6% | [1] |
| Arctic Region | > 66.5° N | | 2.1548 | 4.2% | [1] |

[1] Wilkerson et al. (2010); [2] Hoare (2014); [3] Defined in this study

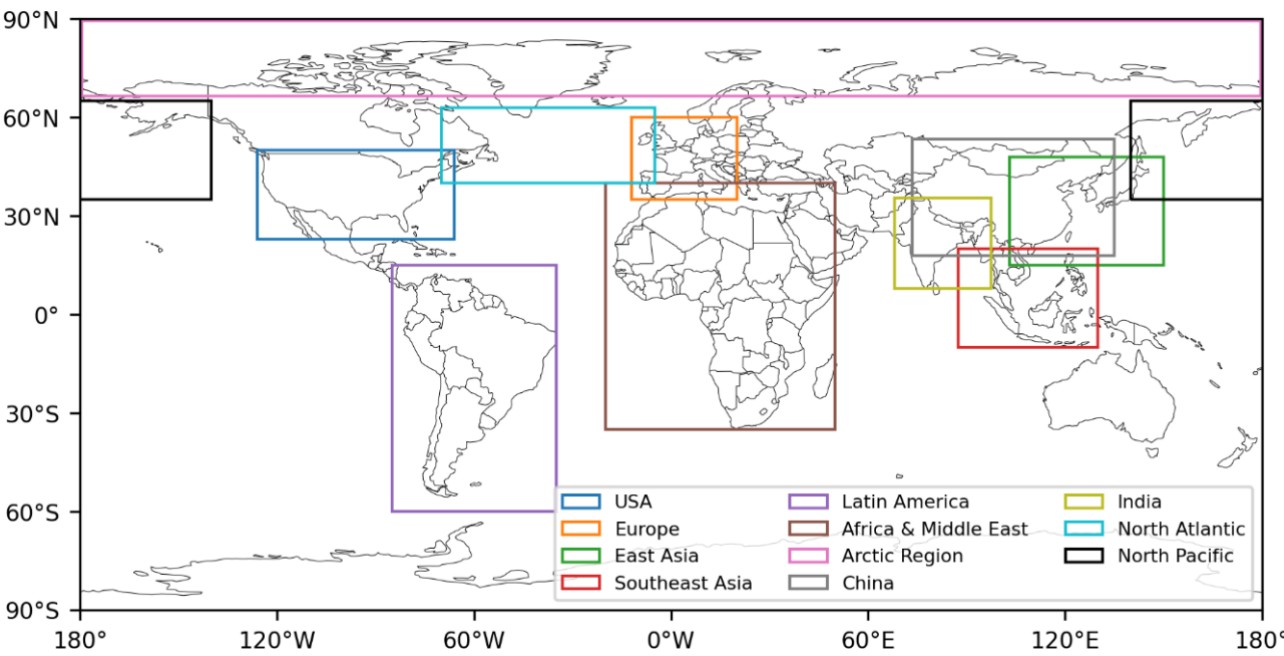

Figure 1: Spatial bounding boxes of the key regions of interest that is used to estimate the regional aviation activity and emissions. The specific dimensions of the bounding boxes are listed in Table 2. Basemap plotted using Cartopy 0.21.1 © Natural Earth; license: public domain.

## 3 Results

We first present the annual air traffic activity and emissions for 2019 (Section 3.1), which is used as a base year to represent normal traffic conditions, followed by quantifying the impacts of COVID-19 on aviation (Section 3.2). Table 3 summarises the global air traffic activity, fuel consumption and emissions from 2019 to 2021. These annual statistics are then compared with existing studies (Section 3.3), followed by an evaluation of granular statistics from individual flights (Section 3.4).

**Table 3: Annual statistics on the global aviation activity, fuel burn and emissions from 2019 to 2021.**

| Annual statistics | 2019 | 2020 | 2021 | % Change 2020 vs. 2019 | % Change 2021 vs. 2019 |
|---|---|---|---|---|---|
| Total number of flights | 40,221,182 | 27,911,214 | 35,576,376 | -31% | -12% |
| - Jet | 33,224,736 | 20,302,177 | 24,458,494 | -39% | -26% |
| - Turboprop | 3,231,103 | 2,719,339 | 3,754,998 | -16% | 16% |
| - Piston | 3,765,343 | 4,889,698 | 7,362,884 | 30% | 96% |
| Distance travelled ($\times 10^9$ km) | 60.94 | 34.50 | 41.90 | -43% | -31% |
| - Jet | 59.00 | 32.59 | 39.16 | -45% | -34% |
| - Turboprop | 1.34 | 1.13 | 1.56 | -15% | 17% |
| - Piston | 0.61 | 0.78 | 1.18 | 29% | 94% |
| Mean passenger load factor (%)* | 83% | 59% | 67% | -29% | -19% |
| Mean aircraft mass (kg) | 64079 | 49593 | 46533 | -23% | -6.2% |
| Fuel burn (Tg) | 283 | 146 | 166 | -48% | -41% |
| Fuel burn per distance (kg km$^{-1}$) | 4.636 | 4.240 | 3.958 | -8.5% | -15% |
| $CO_2$ (Tg) | 893 | 462 | 524 | -48% | -41% |
| $H_2O$ (Tg) | 348 | 180 | 204 | -48% | -41% |
| OC (Gg) | 5.65 | 2.93 | 3.32 | -48% | -41% |
| $SO_2$ (Gg) | 339 | 176 | 199 | -48% | -41% |
| $S^{VI}$ (Gg) | 6.92 | 3.58 | 4.06 | -48% | -41% |
| $NO_x$ (as $NO_2$, Tg) | 4.49 | 2.26 | 2.55 | -50% | -43% |
| CO (Gg) | 400 | 227 | 272 | -43% | -32% |
| HC (Gg) | 33.9 | 20.9 | 25.0 | -38% | -26% |
| nvPM mass (Gg) | 21.4 | 9.93 | 11.0 | -54% | -49% |
| nvPM number ($\times 10^{26}$) | 2.83 | 1.46 | 1.66 | -48% | -41% |
| Mean EI $NO_x$ (g kg$^{-1}$) | 15.9 | 15.4 | 15.4 | -2.8% | -3.2% |
| Mean EI CO (g kg$^{-1}$) | 1.42 | 1.55 | 1.64 | 9.6% | 16% |
| Mean EI HC (g kg$^{-1}$) | 0.120 | 0.143 | 0.151 | 19% | 26% |
| Mean nvPM EI$_m$ (g kg$^{-1}$) | 0.076 | 0.068 | 0.066 | -10.4% | -12% |
| Mean nvPM EI$_n$ ($\times 10^{15}$ kg$^{-1}$) | 1.002 | 0.998 | 1.001 | -0.4% | -0.1% |

*: The passenger load factor for each flight was derived using the global and regional data published by ICAO and IATA (refer to Section 2.2 and the SI §S3).

## 3.1 Annual statistics: 2019

In 2019, there are 40,221,182 unique flights recorded in the ADS-B dataset (83% by jet aircraft, 8% turboprop, and 9% piston aircraft) and the annual flight distance travelled amounts to 60.9 $\times 10^9$ km (97% by jet aircraft, 2% turboprop, and 1% piston). Fig. 2a shows the 2019 global air traffic density, defined as the total flight distance flown divided by the regional surface area and time. 92% of the annual flight distance flown occurred in the Northern Hemisphere, and 63% in the northern mid-latitudes (30°N – 60°N). There were small amounts of flight distance flown in the Arctic Circle above 66.5°N (0.62%) and below 45°S (0.06%). On a regional level, Europe, USA, and East Asia, which cover 7.6% of the global surface area, had the highest air traffic densities (mean of 0.152, 0.116, and 0.059 km$^{-1}$ h$^{-1}$ respectively) and are responsible for a total of 55% of the global

annual flight distance flown (Table 4). The North Atlantic (0.030 km$^{-1}$ h$^{-1}$) and North Pacific flight corridors (0.012 km$^{-1}$ h$^{-1}$) account for 4.9% and 3.9% of the annual distance travelled, respectively, while 52% of the globe has an air traffic density below 0.001 km$^{-1}$ h$^{-1}$.

Global aviation consumed 282 Tg of fuel in 2019, of which ~92% was burnt in the Northern Hemisphere (Fig. S13b) and ~79% above 25,000 feet (Fig. 3a). Air traffic activity within the spatial bounding boxes of the US, Europe, and East Asia accounted for 47% of the annual fuel consumption (Table 4). There are notable differences in the regional distribution of fuel consumption relative to the air traffic density. The proportion of fuel consumption in the US (19%) is lower than its share of aviation activity (27% of the global annual flight distance flown), and the mean fuel consumption per distance flown (3.29 kg km$^{-1}$) is 29% lower than the global average (4.64 kg km$^{-1}$) (Fig. 4 and Table 4). In contrast, the North Atlantic and North Pacific flight corridors have a higher share of fuel consumption (13%) than their distance flown (8.8%), and the mean distance-specific fuel consumption (6.61 kg km$^{-1}$) is 43% higher than the global average (Fig. 4 and Table 4). The discrepancies in distance-specific fuel consumption are due to a higher proportion of short-haul domestic flights in the US predominantly served by smaller narrow-body aircraft, while larger and heavier wide-body aircraft are generally used for long-haul transoceanic flights. We note that the distance-specific fuel consumption in China (4.99 kg km$^{-1}$) is 52% and 21% higher than the US (3.29 kg km$^{-1}$) and Europe (4.14 kg km$^{-1}$), respectively, and this is most likely due to the difference in: (i) airspace structure (Rosenow et al., 2022; IFALPA, 2008) which could cause a higher proportion of flights in Chinese airspace to cruise at lower altitudes of between 25,000 and 35,000 feet (44% of total flight distance flown) relative to other regions (31% of flight distance flown globally) (Fig. S14); and (ii) fleet composition mix (narrow/wide-body aircraft). Global aviation emissions of $CO_2$ (892 Tg in 2019), $H_2O$ (347 Tg), OC (5.65 Gg), $SO_2$ (339 Gg) and $S^{VI}$ (6.92 Gg) are calculated with a constant EI and therefore have the same spatial distribution relative to fuel consumption. The emissions of $NO_X$ (4.49 Tg), CO (400 Gg), HC (33.9 Gg), nvPM mass (21.4 Gg) and number (2.83 $\times 10^{26}$) depend on engine operating conditions and ambient meteorology. Around 76% of annual $NO_X$ emissions were emitted above 25,000 feet, while the proportion of CO (~48%) and HC (~55%) emitted at cruise are notably lower than the fuel consumption (~79%) because these pollutants generally have lower EI's at high engine thrust settings (DuBois and Paynter, 2006; Durdina et al., 2017; EASA, 2021; Lobo et al., 2015; Boies et al., 2015) (Fig. 3). At altitudes above 45,000 feet, the mean nvPM $EI_m$ (0.39 g kg$^{-1}$) and $EI_n$ (4.5 $\times 10^{15}$ kg$^{-1}$) are around 4–5 times larger than the global mean values (0.076 g kg$^{-1}$ and 1.0 $\times 10^{15}$ kg$^{-1}$) (Fig. 3b). This phenomenon can be attributed to a higher prevalence of private business jets with large nvPM $EI_m$ and $EI_n$ of up to 0.58 g kg$^{-1}$ and 7 $\times 10^{15}$ kg$^{-1}$, respectively (Table S8). The mean EI's of CO, HC and nvPM $EI_m$ and $EI_n$ in high air traffic density regions (US and Europe) are ~30% larger than the global average because of a higher proportion of the flight segments in descent, while these EI's are ~36% smaller than the global average over the oceans (North Atlantic and North Pacific) because the engines generally operate at higher thrust settings at cruise (Fig. 4 and Table 4).

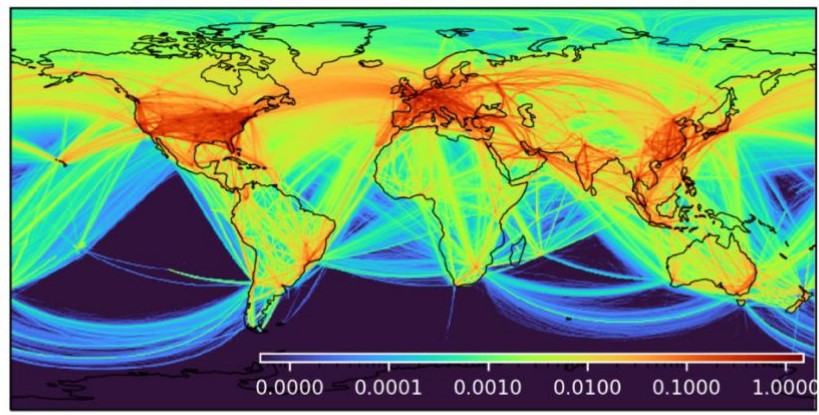

(a) Air traffic density: 2019 $(km^{-1} h^{-1})$

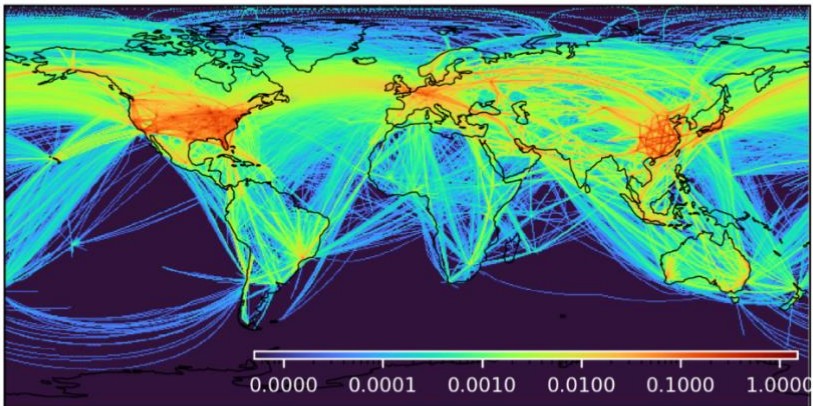

(b) Air traffic density: April-2020 $(km^{-1} h^{-1})$

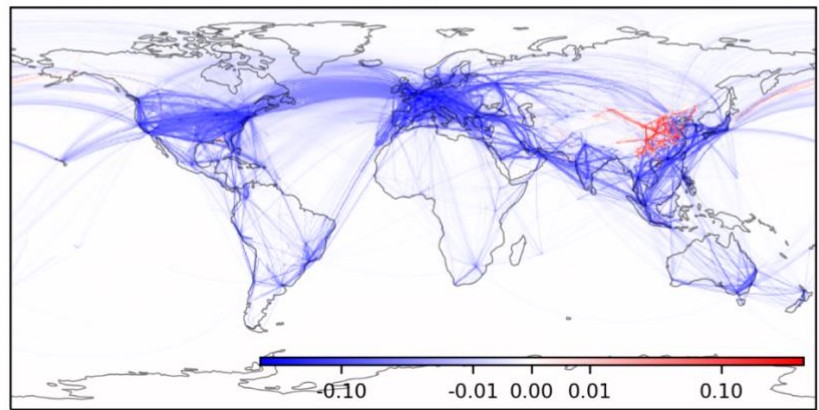

(c) Change in air traffic density: 2019 vs. 2020 $(km^{-1} h^{-1})$

**Figure 2: The global (a) annual air traffic density in 2019; (b) monthly air traffic density in April-2020, where air traffic activity was at a minimum due to the COVID-19 pandemic; and (c) the change in annual air traffic density between 2019 and 2020. Basemap plotted using Cartopy 0.21.1 © Natural Earth; license: public domain.**

## 3.2 Impacts from COVID-19

COVID-19 led to significant reductions in global air traffic activity. The total flight distance travelled reached a minimum in April 2020 (-76% globally relative to April 2019, shown in Fig. 2b), and the annual flight distances flown in 2020 and 2021 are 43% and 31% lower than in 2019, respectively (Table 3). Fig. 2c compares the change in air traffic density between 2019 and 2020 and shows significant regional variability: the largest year-on-year reduction in flight distance travelled was observed in the North Atlantic (-61%), Southeast Asia (-61%) and Europe (-59%), regions, which have a higher proportion of

international flights, followed by Africa and the Middle East (-57%), Latin America (-52%), the North Pacific (-33%) and US (-31%), and East Asia (-24%). East China (25–42°N, 100–118°E) is the only region that recorded air traffic growth in 2020 compared to 2019 (+21%).

**Table 4: Regional aviation activity, fuel consumption and emissions for 2019. The statistics for 2020 and 2021 can be found in Tables S9 and S10 (SI §S5).**

| Regional statistics: 2019 | Global | USA | Europe | East Asia | SEA | Latin America | Africa & Middle East | China | India | North Atlantic | North Pacific | Arctic Region |
|---|---|---|---|---|---|---|---|---|---|---|---|---|
| Distance travelled ($\times 10^9$ km) | 60.92 | 16.30 | 8.858 | 8.304 | 3.989 | 2.250 | 4.631 | 8.946 | 2.551 | 2.975 | 2.387 | 0.382 |
| - Percentage by region[a] | - | 27% | 15% | 14% | 6.5% | 3.7% | 7.6% | 15% | 4.2% | 4.9% | 3.9% | 0.6% |
| Air traffic density (km$^{-1}$ h$^{-1}$)[b] | 0.014 | 0.116 | 0.152 | 0.059 | 0.029 | 0.006 | 0.009 | 0.047 | 0.032 | 0.030 | 0.012 | 0.002 |
| Fuel burn (Tg) | 282 | 53.7 | 36.7 | 41.7 | 19.6 | 9.5 | 23.3 | 44.7 | 13.2 | 18.8 | 16.7 | 3.00 |
| - Percentage by region[a] | - | 19% | 13% | 15% | 6.9% | 3.4% | 8.2% | 16% | 4.7% | 6.7% | 5.9% | 1.1% |
| Fuel burn per distance (kg km$^{-1}$) | 4.636 | 3.293 | 4.138 | 5.025 | 4.906 | 4.232 | 5.029 | 4.993 | 5.155 | 6.313 | 6.988 | 7.836 |
| $CO_2$ (Tg) | 892 | 170 | 116 | 132 | 61.8 | 30.1 | 73.6 | 141 | 41.5 | 59.3 | 52.7 | 9.46 |
| $H_2O$ (Tg) | 347 | 66.0 | 45.1 | 51.3 | 24.1 | 11.7 | 28.6 | 54.9 | 16.2 | 23.1 | 20.5 | 3.68 |
| OC (Gg) | 5.65 | 1.07 | 0.733 | 0.835 | 0.391 | 0.190 | 0.466 | 0.893 | 0.263 | 0.376 | 0.334 | 0.060 |
| $SO_2$ (Gg) | 339 | 64.4 | 44.0 | 50.1 | 23.5 | 11.4 | 27.9 | 53.6 | 15.8 | 22.5 | 20.0 | 3.59 |
| $S^{VI}$ (Gg) | 6.92 | 1.31 | 0.898 | 1.02 | 0.479 | 0.233 | 0.570 | 1.09 | 0.322 | 0.460 | 0.408 | 0.073 |
| $NO_x$ (as $NO_2$, Tg) | 4.49 | 0.756 | 0.550 | 0.686 | 0.327 | 0.147 | 0.375 | 0.722 | 0.221 | 0.300 | 0.303 | 0.055 |
| - Percentage by region[a] | - | 17% | 12% | 15% | 7.3% | 3.3% | 8.4% | 16% | 4.9% | 6.7% | 6.7% | 1.2% |
| CO (Gg) | 400 | 104 | 74.9 | 57.8 | 29.6 | 14.5 | 30.6 | 58.4 | 15.7 | 11.3 | 15.2 | 1.39 |
| - Percentage by region[a] | - | 26% | 19% | 14% | 7.4% | 3.6% | 7.7% | 15% | 3.9% | 2.8% | 3.8% | 0.3% |
| HC (Gg) | 33.9 | 9.41 | 5.71 | 4.51 | 2.19 | 1.01 | 2.57 | 4.58 | 1.26 | 1.30 | 1.47 | 0.164 |
| - Percentage by region[a] | - | 28% | 17% | 13% | 6.5% | 3.0% | 7.6% | 14% | 3.7% | 3.8% | 4.3% | 0.5% |
| nvPM mass (Gg) | 21.4 | 5.18 | 2.95 | 3.33 | 1.55 | 0.770 | 1.63 | 3.49 | 0.970 | 1.113 | 0.851 | 0.114 |
| - Percentage by region[a] | - | 24% | 14% | 16% | 7.2% | 3.6% | 7.6% | 16% | 4.5% | 5.2% | 4.0% | 0.5% |
| nvPM number ($\times 10^{26}$) | 2.827 | 0.730 | 0.423 | 0.452 | 0.192 | 0.099 | 0.206 | 0.484 | 0.136 | 0.117 | 0.108 | 0.012 |
| - Percentage by region[a] | - | 26% | 15% | 16% | 6.8% | 3.5% | 7.3% | 17% | 4.8% | 4.1% | 3.8% | 0.4% |
| Mean EI $NO_x$ (g kg$^{-1}$) | 15.9 | 14.1 | 15.0 | 16.4 | 16.7 | 15.5 | 16.1 | 16.2 | 16.8 | 16.0 | 18.1 | 18.2 |
| Mean EI CO (g kg$^{-1}$) | 1.42 | 1.94 | 2.04 | 1.39 | 1.51 | 1.52 | 1.31 | 1.31 | 1.19 | 0.60 | 0.91 | 0.46 |
| Mean EI HC (g kg$^{-1}$) | 0.120 | 0.175 | 0.156 | 0.108 | 0.112 | 0.106 | 0.110 | 0.103 | 0.096 | 0.069 | 0.088 | 0.055 |
| Mean nvPM $EI_m$ (g kg$^{-1}$) | 0.076 | 0.097 | 0.080 | 0.080 | 0.079 | 0.081 | 0.070 | 0.078 | 0.074 | 0.059 | 0.051 | 0.038 |
| Mean nvPM $EI_n$ ($\times 10^{15}$ kg$^{-1}$) | 1.001 | 1.361 | 1.155 | 1.082 | 0.983 | 1.035 | 0.883 | 1.083 | 1.036 | 0.621 | 0.649 | 0.406 |

[a]: The percentages of each region do not add up to 100% because there are some overlapping between the regional bounding boxes; and when taken together, these regions do not cover 100% of Earth's surface area (refer to Fig. 1 and Table 2).

[b]: The air traffic density (ATD) is defined as the total flight distance flown in the region divided by its surface area and time,

$$\text{ATD } [\text{km}^{-1}\,\text{h}^{-1}] = \frac{\sum \text{Annual flight distance flown [km]}}{\text{Surface area } [\text{km}^2] \times (365 \times 24\,[\text{h}])}.$$

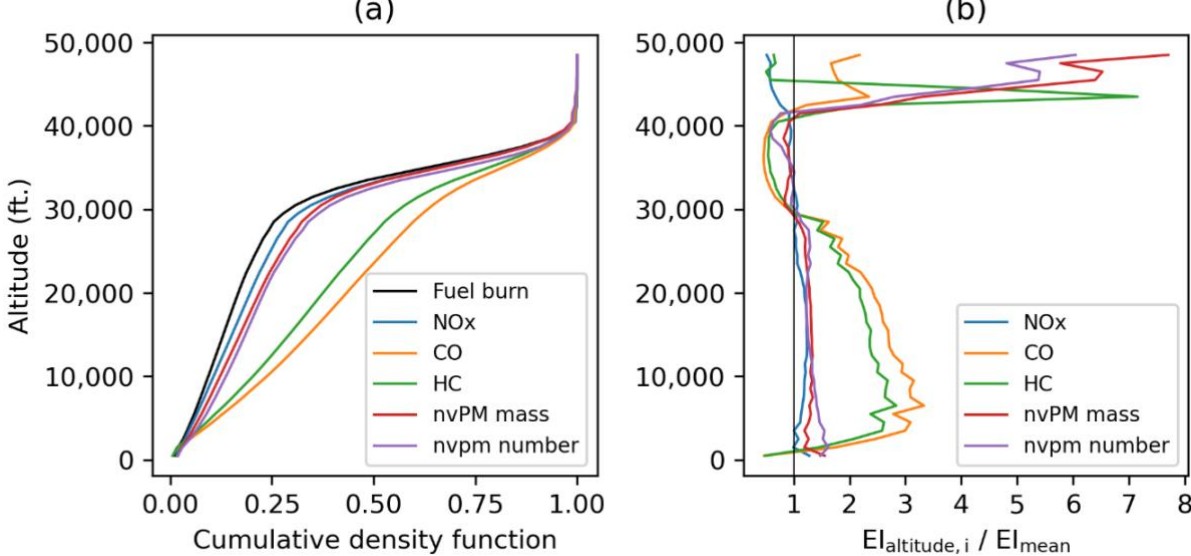

**Figure 3: (a) Cumulative density function of the 2019 annual fuel burn by altitude, and (b) the ratio of emission indices (EI) for nitrogen oxide (NO$_X$, in blue), carbon monoxide (CO, in orange), hydrocarbons (HC, in green) and nvPM mass (in red) and number (in purple) at different altitudes relative to their respective global mean EI's.**

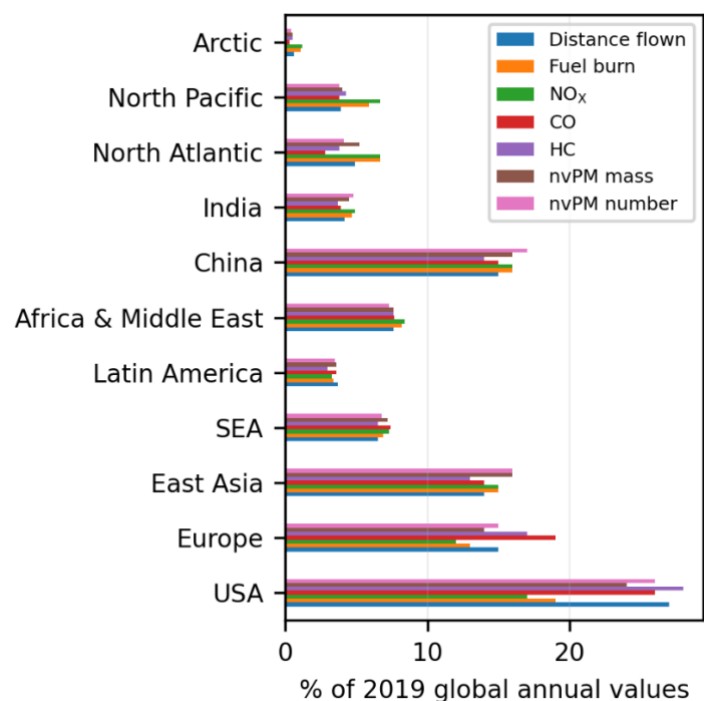

**Figure 4: Percentage breakdown of the 2019 global annual flight distance flown, fuel consumption, NO$_X$, CO, HC, nvPM mass and number emissions by region.**

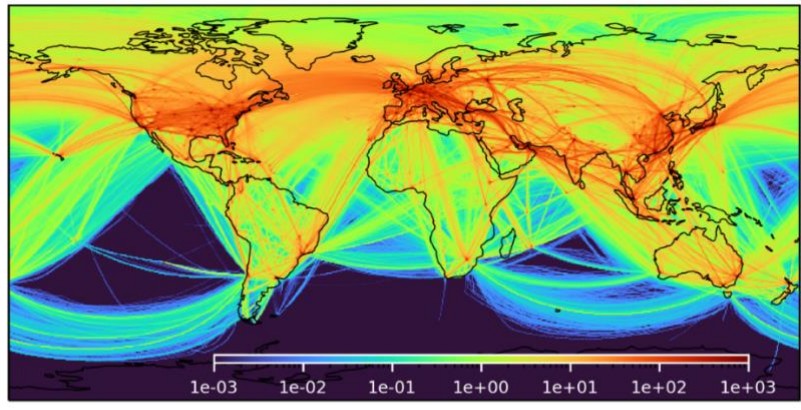

(a) 2019 NO$_X$ emissions (kg km$^{-2}$): GAIA

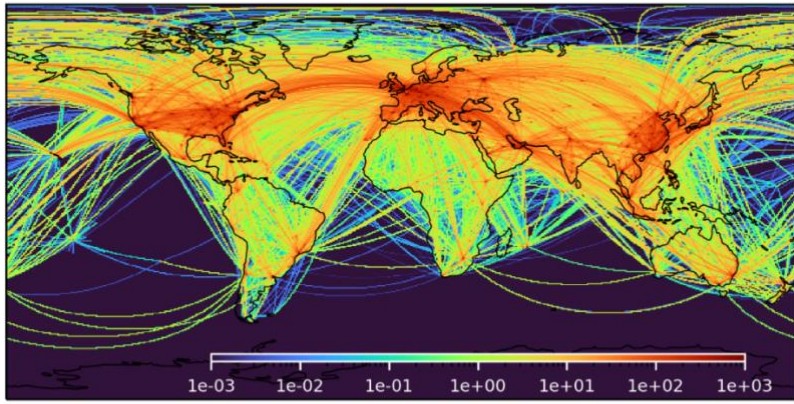

(b) 2019 NO$_X$ emissions (kg km$^{-2}$): Quadros et al. (2022)

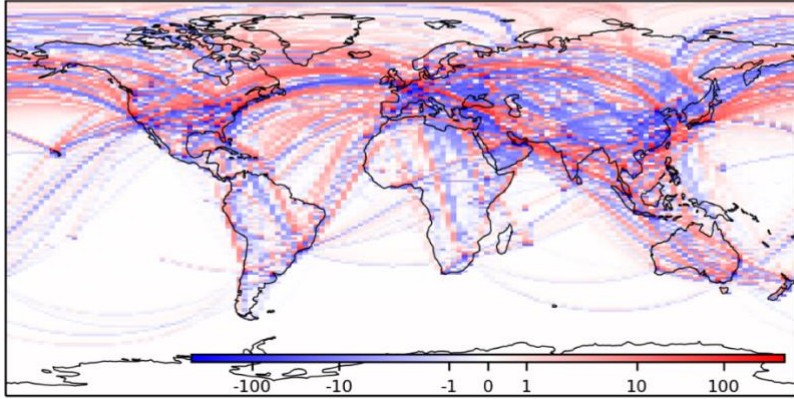

(c) Difference (kg km$^{-2}$): GAIA - Quadros et al. (2022)

**Figure 5: Spatial distribution of the 2019 column-integrated NO$_X$ emissions from (a) GAIA with actual flight trajectories versus (b) estimates from Quadros et al. (2022) which used monthly-averaged flight trajectories, and (c) the absolute difference in column-integrated NO$_X$ emissions between (a) and (b). Basemap plotted using Cartopy 0.21.1 © Natural Earth; license: public domain.**

Globally, reductions in the annual fuel consumption and $CO_2$ emissions (-48% in 2020 and -41% in 2021 relative to 2019 levels) are greater than the change in flight distance travelled (-43% and -31% respectively), thereby causing the mean fuel consumption per flight distance flown in 2020 (4.24 kg km$^{-1}$) and 2021 (3.96 kg km$^{-1}$) to be 9% and 15% lower than 2019 levels (4.64 kg km$^{-1}$), respectively (Table 3). The lower distance-specific fuel consumption is most likely caused by the lower global annual mean aircraft mass (-23% in 2020 and -6.2% in 2021 relative to 2019, Table 3), which in turn, can be attributed to the: (i) lower annual mean passenger load factor (59% in 2020 and 67% in 2021 vs. 83% in 2019); (ii) reduction in long-haul flights (> 6 h) in 2020 and 2021 that are predominantly flown by wide-body aircraft (3% of all flights) relative to 2019 (5%), where their mean aircraft mass is around 2–4 times larger than short- and medium-haul flights (< 6 h) that are generally flown by narrow-body aircraft (Tables 5, S11 and S12); and (iii) increased usage of private jets in the global fleet composition (Fig. S9). The change in aircraft fleet composition and mission profile during the COVID-19 pandemic also contributed to differences in the mean EI's of CO (+13% relative to 2019), HC (+23%), $NO_X$ (-3.0%) and nvPM $EI_m$ (-12%) (Table 3).

### 3.3 Comparison with other studies

GAIA's 2019 estimate of aviation's total fuel consumption (283 Tg) is 12% lower than the IEA's top-down estimate of the global jet kerosene consumption (320 Tg) (IEA, 2020) and 4.7% lower than bottom-up estimates from Quadros et al. (2022) (297 Tg, derived from Flightradar24 ADS-B telemetry). The incomplete coverage area of ADS-B receiver networks (Fig. S1) is likely one of the limitations causing the annual fuel consumption estimates from both GAIA and Quadros et al. (2022) to be lower than IEA's top-down estimates. Several factors contribute to the differences in the annual fuel consumption between GAIA and Quadros et al. (2022): (i) fuel consumption from ground operations (i.e., taxi, take-off, and use of APUs) are not included in GAIA; (ii) use of actual flight trajectories and meteorology in GAIA, which comes at a cost where ~27% of flights in 2019 have incomplete trajectories (i.e., waypoints do not begin and end at the origin-destination airports respectively) (Fig. S7d); and (iii) methodological differences where Quadros et al. (2022) used monthly-averaged flight trajectories and wind fields and applied a constant scaling factor to account for routing inefficiencies (40.5 NM + 3.87% of the great circle distance). The International Council on Clean Transportation (ICCT) inventory study estimates the 2019 annual $CO_2$ emissions from all commercial operations to be 918 Tg (Graver et al., 2020), and our estimates (892 Tg of annual $CO_2$ emissions from jet and turboprop aircraft) differ by 2.8%. We also compare the relative change in global aviation $CO_2$ emissions between 2019 and 2020, and our estimates are in good agreement with Quadros et al. (2022) (-48.2% in this study vs. -47.1%) and Liu et al. (2020) (-44.0% vs. -43.9% for the first half of 2020).

The 2019 mean EI $NO_X$ from GAIA (15.9 g kg$^{-1}$) is within 2% when compared with Quadros et al. (2022) (15.6 g kg$^{-1}$). It is also 17% and 11% larger, relative to the 2002 AERO2K (13.2 g kg$^{-1}$) and 2006 (14.2 g kg$^{-1}$) AEDT global aviation emission inventories (Eyers et al., 2005; Wilkerson et al., 2010), respectively, which likely reflects the increasing use of more fuel-efficient engines that operate at higher combustion temperatures and pressures (Kyprianidis and Dahlquist, 2017; Freeman et al., 2018). The emissions profiles for $NO_X$, CO, HC, and nvPM vary for different aircraft-engine types (Fig. S18 to S22), and

differences in the treatment of aircraft-engine combination between our study and Quadros et al. (2022) could also contribute

to discrepancies in the 2019 EI CO (1.4 g kg$^{-1}$ from GAIA vs. 2.7 g kg$^{-1}$, -48%), HC (0.12 vs. 0.14 g kg$^{-1}$, -14%), nvPM EI$_m$ (76 mg kg$^{-1}$ vs. 33 mg kg$^{-1}$, +130%) and EI$_n$ (1.00 vs. 1.17 ×10$^{15}$ kg$^{-1}$, -14.5%). The significantly lower estimate of nvPM EI$_m$ from Quadros et al. (2022) (33 mg kg$^{-1}$) relative to this study (76 mg kg$^{-1}$) could also be attributed to their use of the Döpelheuer & Lecht relation to scale the nvPM emissions from ground to cruise (Peck et al., 2013; Döpelheuer and Lecht, 1998), which was developed based on limited number of cruise measurements and could underestimate the nvPM EI$_m$ (Abrahamson et al.,

2016).

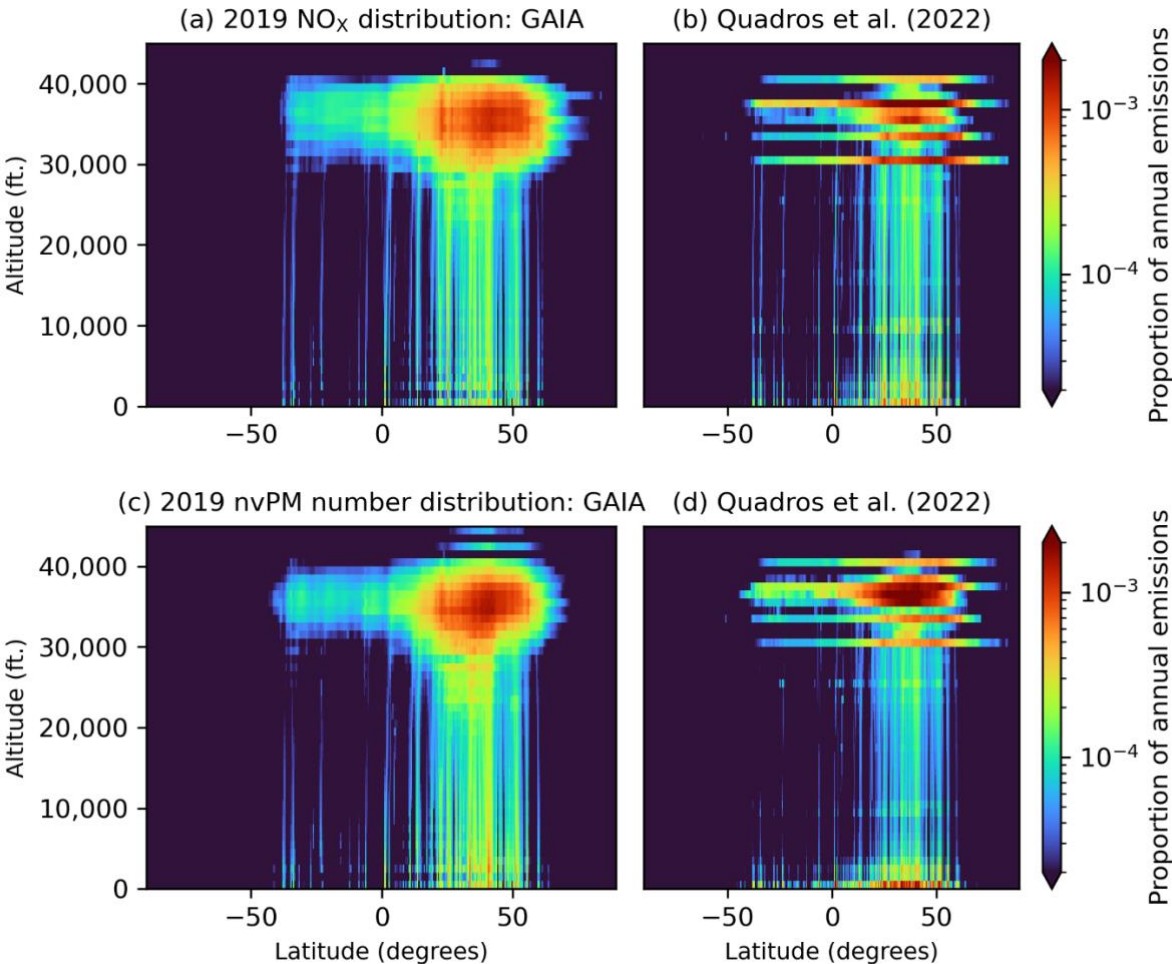

**Figure 6: Distribution of the 2019 NO$_X$ and nvPM number emissions by latitude and altitude from GAIA with actual flight trajectories (subplots a and c) versus estimates from Quadros et al. (2022) with monthly-averaged flight trajectories (subplots b and d). The colour bar represents the proportion of annual emissions, i.e., annual emissions at 360 each grid cell divided by the global annual emissions, where the summation of values across all grid cells would be equal to 1.**

Notably, there are significant differences in the spatial distribution of global aviation emissions between GAIA and Quadros et al. (2022) (Fig. 5, 6 and S15): the use of actual flight trajectories in GAIA caused the fuel consumption and pollutants to be more spatially dispersed; while Quadros et al. (2022) used monthly-averaged flight trajectories, resulting in emissions that are more concentrated along established flight corridors and selected altitudes. The air traffic activity in GAIA is also more representative of real-world operations, where aircraft actively avoid flying over the Himalayas/Tibet and conflict zones (Safe Airspace, n.d.) (i.e., Ukraine, Syria, Yemen, Libya, and North Korea) (Fig. 5).

### 3.4 Flight-level statistics

We categorise the 2019 global air traffic activity into three groups based on their flight duration (Wilkerson et al., 2010), consisting of short- ($t \leq 3$ h), medium- ($3 < t < 6$ h), and long-haul ($t > 6$ h) flights (Table 5). In 2019, short-haul flights (83% of all flights and 49% of the total distance travelled) accounted for 35% of the total $CO_2$ emissions, and the largest fraction of CO (59%), HC (52%) and nvPM mass (45%) and number (51%) emissions because a greater proportion of flight time is in descent where the EI's of these pollutants are generally at its maximum (Fig. S19 to S22). In contrast, long-haul flights (5% of all flights and 26% of the distance travelled) are responsible for the largest share of $CO_2$ (43%) and $NO_X$ (49%) emissions because of the significantly larger mean aircraft mass (~3.5 times greater than short-haul flights) that can be attributed to the use of larger wide-body aircraft and higher fuel mass fraction (~26% of the initial aircraft mass vs. ~6% for short-haul flights) (Table 5 and Fig. S9d). Travel restrictions due to COVID-19 has caused the proportion of international flights to decrease from 39% (2019) to 23% (2020 and 2021). As a result, the share of short-haul flights increased from 83% to 88% while long-haul flights decreased from 5% to 3%.

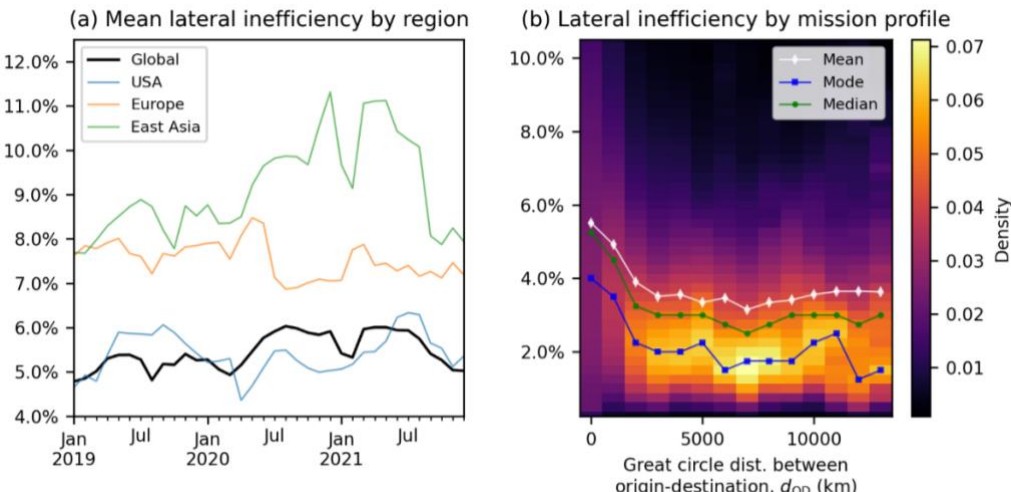

**Figure 7: The (a) mean lateral inefficiency of flights in different spatial domains (global, USA, Europe, and East Asia); and (b) distribution of lateral inefficiency when flights are categorised by mission profile, defined by the great circle distance between their respective origin and destination airports. These statistics excludes the 1.5% of flights, where their erroneous trajectories were replaced with the great-circle path between the origin-destination airport (SI §S1.2).**

**Table 5: Breakdown of aviation activity, fuel consumption and emissions for 2019 by flight duration. The statistics for 2020 and 2021 can be found in Tables S11 and S12 (SI §S5).**

| Flight-level statistics: 2019 | All flights | Short-haul ($t \leq 3h$) | | Medium-haul ($3 < t \leq 6$) | | Long-haul ($t > 6$) | |
|---|---|---|---|---|---|---|---|
| | | Value | % total | Value | % total | Value | % total |
| Number of flights | 40,221,182 | 33,245,965 | 82.7% | 4,987,165 | 12.4% | 1,988,052 | 4.9% |
| Number of night flights[a] | 7,120,502 | 5,723,673 | 80.4% | 1,098,494 | 15.4% | 298,335 | 4.2% |
| Distance travelled (x$10^9$ km) | 60.94 | 29.80 | 48.9% | 15.61 | 25.6% | 15.54 | 25.5% |
| Fuel burn (Tg) | 283 | 97.8 | 34.6% | 63.6 | 22.5% | 121 | 42.9% |
| Fuel burn per dist. (kg km$^{-1}$) | 4.636 | 3.281 | - | 4.074 | - | 7.799 | - |
| Mean flight time (h) | 2.06 | 1.35 | - | 4.00 | - | 9.05 | - |
| Mean flight length (km) | 1515 | 896 | - | 3130 | - | 7817 | - |
| Mean aircraft mass (kg) | 64079 | 50643 | - | 90590 | - | 221298 | - |
| - Fuel fraction[b] | 7.85% | 5.74% | - | 14.7% | - | 25.8% | - |
| $CO_2$ (Tg) | 892 | 309 | 34.6% | 201 | 22.5% | 383 | 42.9% |
| $NO_X$ (as $NO_2$, Tg) | 4.49 | 1.35 | 30.1% | 0.92 | 20.4% | 2.22 | 49.4% |
| CO (Gg) | 400 | 233 | 58.3% | 78.8 | 19.7% | 87.7 | 21.9% |
| HC (Gg) | 33.9 | 17.6 | 51.9% | 6.87 | 20.3% | 9.38 | 27.7% |
| nvPM mass (Gg) | 21.4 | 9.54 | 44.5% | 5.67 | 26.4% | 6.23 | 29.1% |
| nvPM number (x$10^{26}$) | 2.83 | 1.45 | 51.2% | 0.782 | 27.6% | 0.594 | 21.0% |
| Mean EI $NO_X$ (g kg$^{-1}$) | 15.89 | 13.81 | - | 14.43 | - | 18.32 | - |
| Mean EI CO (g kg$^{-1}$) | 1.42 | 2.38 | - | 1.24 | - | 0.72 | - |
| Mean EI HC (g kg$^{-1}$) | 0.12 | 0.18 | - | 0.11 | - | 0.08 | - |
| Mean nvPM EI$_m$ (g kg$^{-1}$) | 0.076 | 0.098 | - | 0.089 | - | 0.051 | - |
| Mean nvPM EI$_n$ (x$10^{15}$ kg$^{-1}$) | 1.002 | 1.483 | - | 1.230 | - | 0.490 | - |

[a]: Night flights are identified when their mean solar direct radiation (SDR) throughout their flight trajectory is < 1 W m$^{-2}$.
[b]: Fuel fraction = total fuel mass/initial aircraft mass

The global mean lateral route inefficiency, i.e., additional distance flown relative to the great circle distance between the origin-destination airport ($d_{OD}$), increased from 5.2% in 2019 to 5.6% in 2020–21 with regional variations (Fig. 7a): reductions in air traffic levels due to COVID-19 likely led to small improvements in the mean lateral inefficiency in the US (5.5% in 2019 vs. 5.1% in 2020) and Europe (7.7% vs. 7.5%); while an increase in lateral inefficiency in East Asia (from 8.3% to 9.5%) is likely caused by the air traffic growth experienced in China (Fig. 2c). Flights in East Asia/China tend to follow pre-defined air traffic corridors relative to a more direct routing in the US (Fig. 2) because around ~70% of the eastern Chinese airspace is restricted for military usage relative to ~15% in the US (Rosenow et al., 2022), and this is likely the primary factor causing the mean lateral inefficiency in East Asia (9.1%) to be around two times higher than the US (5.4%). Our estimate of the mean lateral inefficiency in Europe (~7.5%) is around 2 times larger than the official statistics from EUROCONTROL (2022) (~4%) because the latter computes the lateral inefficiency relative to the shortest constraint route (i.e., constraints imposed by the airspace structure and availability) instead of the great-circle distance. Given the significant regional variability in lateral inefficiency factor, earlier studies that assume a constant lateral inefficiency factor of around 4–6% could: (i) underestimate the total distance travelled for flights outside the US; and (ii) overestimate the flight distance for medium- and long-haul flights (Simone et al., 2013; Quadros et al., 2022).

We group each flight in 2019 by their origin-destination (OD) airport pairs to evaluate the differences in their mean: (i) historical flight distance flown ($d_{GAIA}$) versus $d_{OD}$; and (ii) simulated fuel consumption from GAIA ($f_{GAIA}$) versus the fuel consumption from the great circle trajectory that is estimated from the European Environment Agency emissions calculator ($f_{EEA,OD}$), where $f_{EEA,OD}$ at climb, cruise, and descent (CCD) for each flight is estimated as a function of $d_{OD}$ and aircraft type (European Environment Agency, 2019). These OD statistics have been made open source and are available, as described in the Data Availability statement. In general, there is an inverse relationship between the lateral inefficiency and $d_{OD}$, where the mean lateral inefficiency is 5.1 [0.7, 10.8] % (5th and 95th percentile) for flights with $d_{OD} < 1000$ km, 4.4 [0.9, 10.2] % when $d_{OD}$ is between 1000 and 2000 km, and 2.9 [0.8, 8.6] % for $d_{OD} > 2000$ km (Fig. 7b and Fig. 8a). Our analysis among OD airport pairs also suggests that: (i) the variability of $f_{GAIA}/f_{EEA,OD}$ (1.14 [0.997, 1.35]) is greater than $d_{GAIA}/d_{OD}$ (1.06 [1.01, 1.16]) (Fig. S15); (ii) $d_{GAIA}/d_{OD}$ is relatively symmetrical between OD airport pairs irrespective of direction travelled (Fig. 8a); and (iii) there is a directional bias in $f_{GAIA}/f_{EEA,OD}$ among OD airport pairs, e.g., westbound transatlantic flights to consume more fuel than eastbound transatlantic flights (Fig. 8b), because $f_{GAIA}$ captures the effects of ambient wind patterns.

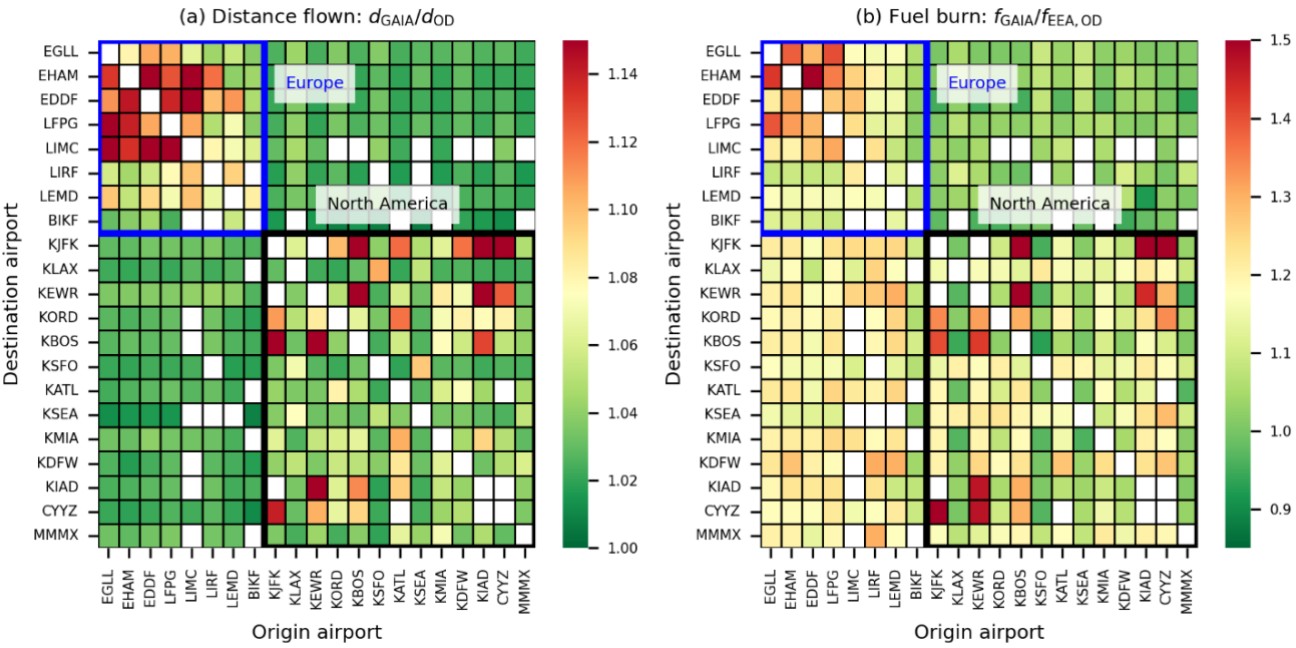

Figure 8: The mean ratio of: (a) the actual flight distance flown from GAIA over the great circle distance ($d_{GAIA}/d_{OD}$); and (b) the simulated fuel consumption from GAIA over the estimated fuel consumption from the great circle trajectory ($f_{GAIA}/f_{EEA,OD}$) across all flights that traversed between major European and North American airport pairs in 2019.

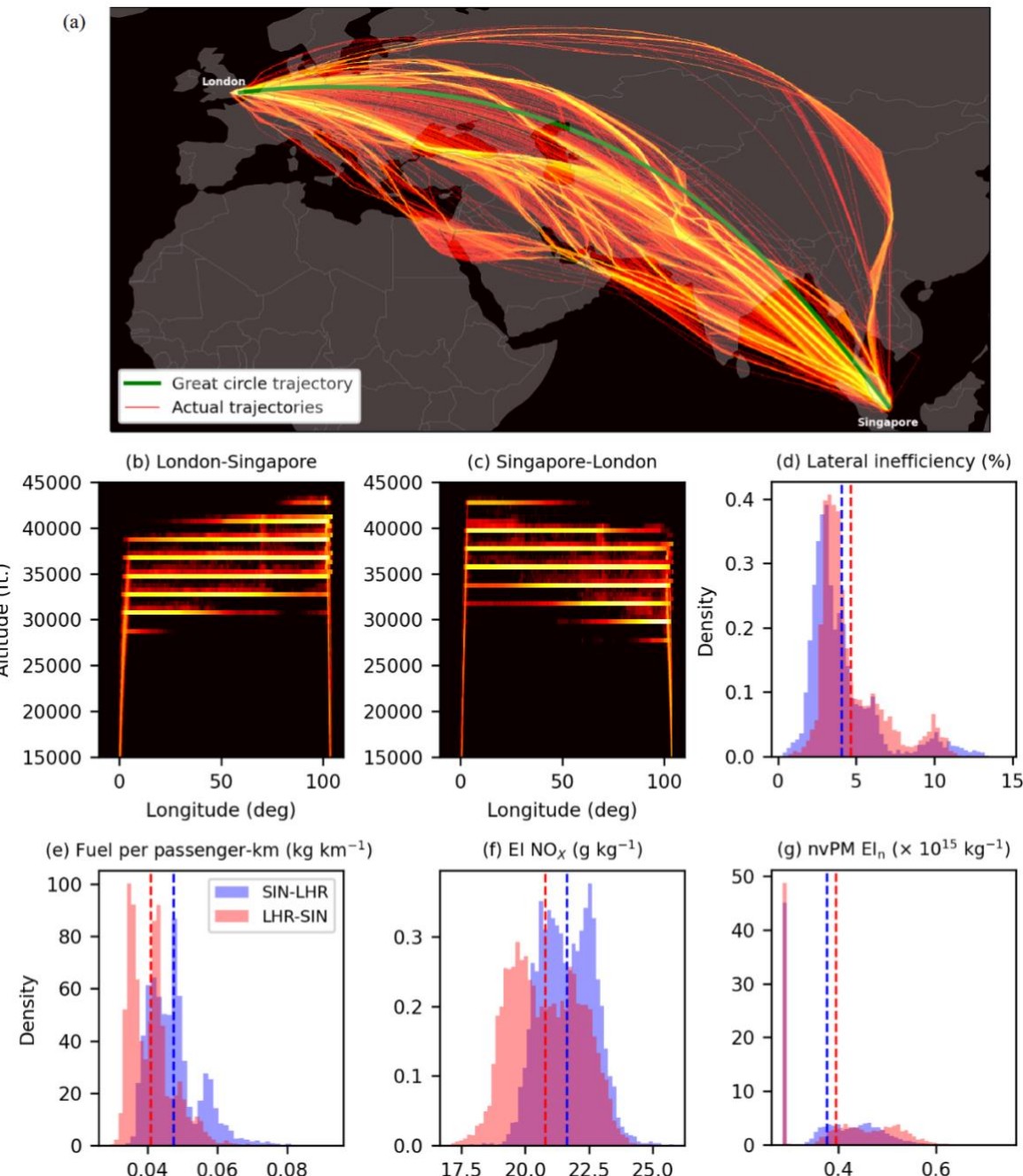

**Figure 9: The (a) lateral and (b – c) vertical trajectory that is flown by flights between London Heathrow Airport (LHR) and Singapore Changi Airport (SIN) between 2019 and 2021 (n = 8705), and probability density function of the (d) lateral inefficiency, (e) fuel consumption per passenger-km; (f) EI NO$_X$; and (g) nvPM EI$_n$ for westbound (SIN–LHR, in blue) and eastbound flights (LHR–SIN, in red). The methodology to estimate the fuel consumption per passenger-km is described in the SI §S5.2. Basemap plotted using Cartopy 0.21.1 © Natural Earth; license: public domain.**

Fig. 9 shows the distribution in the flight trajectory, fuel consumption, and emissions for trips between London Heathrow (LHR) and Singapore Changi Airport (SIN). There are large variabilities in the: (i) flight trajectory profile, where the lateral inefficiency and cruising altitudes range between 2.1–9.9 % (5th and 95th percentile) and 28000–42000 feet respectively; (ii) fuel consumption per passenger-km (0.03–0.06 kg km$^{-1}$, 5th and 95th percentile) arising from differences in aircraft type, passenger LF, and wind conditions; and (iii) mean EI $NO_X$ (19.1–23.1 g kg$^{-1}$) and nvPM $EI_n$ (2.9–5.4 $\times 10^{14}$ kg$^{-1}$), which can be attributed to differences in engine combustor type, thrust settings, and the ambient temperature and humidity. The mean fuel consumption per passenger-km for westbound flights (SIN–LHR: 0.047 kg km$^{-1}$) is 15% larger than eastbound flights (LHR–SIN: 0.041 kg km$^{-1}$) because of the jet stream patterns that tend to flow from west to east. The difference in lateral inefficiency for SIN–LHR (4.1 [1.9, 10.0] %) vs. LHR–SIN (4.7 [2.7, 9.8] %) suggests that: (i) eastbound flights tend to fly more consistent trajectories than westbound flights, and the route optimiser is less sensitive to the flight distance flown because of favourable jet stream conditions; while (ii) westbound flights have a larger variability in the distance flown, suggesting that flights are more actively re-routing where possible to avoid regions with the strongest headwinds. Westbound flights also tend to operate at higher engine thrust settings to counteract the effects of stronger headwinds, thereby leading to a larger mean EI $NO_X$ (+4.1%) and lower nvPM $EI_n$ (-4.7%) respectively relative to eastbound flights.

## 4 Conclusions

In this paper, we developed a high-resolution Global Aviation emissions Inventory based on ADS-B (GAIA) for 2019–2021 using: (i) historical flight trajectories that are derived from ADS-B telemetry data collected by Spire Aviation; (ii) reanalysis weather data from the ECMWF ERA5 HRES; and (iii) existing datasets and methodologies to estimate the fuel consumption, $CO_2$, $NO_X$, CO, HC, OC, $H_2O$, $SO_2$, $S^{VI}$, and nvPM mass and number emissions for each flight. GAIA captures 103.7 million unique flights over three-years, of which 75.2% are completed using jet aircraft and the remainder by turboprop (9.4%) and piston aircraft (15.4%).

In 2019, there were ~40.2 million flights that collectively consumed 283 Tg of fuel to travel 60.9 $\times 10^9$ km (mean distance-specific fuel consumption of 4.64 kg km$^{-1}$), producing 893 Tg of $CO_2$, 348 Tg of $H_2O$, 5.7 Gg of OC, 339 Gg of $SO_2$, 6.9 Gg of $S^{VI}$, 4.5 Tg of $NO_X$, 400 Gg of CO, 34 Gg of unburnt HC, 21.4 Gg of nvPM mass and 2.8 $\times 10^{26}$ nvPM particles. Around 92% of annual fuel consumption occurred in the Northern Hemisphere and 79% of it was burnt at cruise altitudes (> 25,000 feet). The United States, Europe, and East Asia, covering ~8% of the global surface area, accounted for 47% of the fuel consumption and $CO_2$ emissions in 2019. Our estimated global annual fuel consumption for 2019 (283 Tg) is around 12% smaller than top-down estimates from the IEA (320 Tg), and 5% smaller than bottom-up estimates from Quadros et al. (2022) (297 Tg) which used monthly-averaged flight trajectories and wind fields. Globally, COVID-19 led to significant reductions in the annual flight distance flown (-43%) and fuel consumption (-48%) in 2020, with the largest reductions occurring in the North Atlantic (-61% in flight distance and -64% in fuel consumption), Southeast Asia (-61% for both metrics), and Europe (-

59% and -60% respectively), while East China recorded growth in both metrics (+21% and +9% respectively). In 2021, the global annual flight distance and fuel consumption remained 31% and 41% below 2019 levels.

An analysis of individual flight trajectories shows that 83% of all flights in 2019 have durations below 3 h, and these short-haul flights only accounted for 35% of the annual $CO_2$ emissions. In contrast, only 5% of all flights in 2019 are long-haul (duration > 6 h) but they have a disproportionate impact on the annual $CO_2$ (43%) and $NO_X$ (49%) emissions. The mean lateral inefficiency increased from 5.2% (2019) to 5.6% (2020 and 2021) globally, and this is most likely due to a higher proportion of short-haul flights in 2020 and 2021 (~88% of all flights) relative to 2019 (83%) which tend to have a higher routing

inefficiency relative to long-haul flights (Fig. 7b). Regionally, there are small improvements in the mean lateral inefficiency over the US (from 5.5% in 2019 to 5.1% in 2020) and Europe (from 7.7% to 7.5%). In contrast, East Asia experienced an increase in mean lateral inefficiency (from 8.3% in 2019 to 9.5% in 2020), which is likely caused by the growth in air traffic over China. We evaluated 8,649 unique flights between London and Singapore and the trajectories flown, fuel consumption, EI $NO_X$ and nvPM $EI_n$ showed large variabilities between flights, and the fuel consumption and emission indices between east-

and westbound flights can differ by up to 15%. By using flight trajectories that are representative of real-world operations, GAIA improves upon the spatiotemporal distribution of aviation emissions when compared with existing aviation emissions inventories. Future research can utilise GAIA to: (i) quantify the climate impacts from aviation $NO_X$ emissions and contrail cirrus, which radiative forcing have a strong spatiotemporal dependence; and (ii) perform inter-model comparison studies and uncertainty quantification of the climate impacts that arise from these aviation non-$CO_2$ pollutants.

**Supplement**

The supplement related to this article is available online at: <URL>

**Author contributions**

RT and MEJS conceptualised the study, developed the methodology and undertook the investigation. MEJS and MS acquired the ADS-B air traffic dataset. LD acquired the fleet database. RT, ZE and MS were responsible for software development and

480 data curation. RT and MEJS created or sourced the figures. RT wrote the original manuscript. RT, MS, LD and MEJS reviewed and edited the manuscript. MEJS and MS acquired funding. All authors have read and agreed upon the published version of the paper.

**Acknowledgements**

The authors thank Google for providing cloud credits on the Google Cloud Platform, which were utilised to produce the global

aviation emissions inventory and the analysis of results published in this paper. We thank Andreas Schäfer for his efforts in

acquiring funding to purchase the fleet database, and Thomas Dean for reviewing an earlier version of the paper. We thank members of the German Aerospace Center (Christiane Voigt, Daniel Sauer, and Tobias Schripp) for providing in-situ measurements from the ECLIF II/ND-MAX experimental campaign which were used to improve the methodology to estimate the cruise nvPM emissions.

**Data availability**

The following datasets are available on Zenodo: (i) low-resolution gridded outputs from GAIA for 2019–2021 (0.5° longitude × 0.5° latitude, at altitude intervals of 1000 feet, and at a monthly temporal resolution) (https://doi.org/10.5281/zenodo.7969631); (ii) high-resolution gridded outputs from GAIA (0.05° longitude × 0.05° latitude, at altitude intervals of 100 m, and at an hourly temporal resolution) for the full year of 2019 (doi in preparation) and bi-monthly

for 2020 and 2021 (doi in preparation); and (iii) lateral inefficiency and fuel consumption statistics on the origin-destination airport and country pairs for 2019 (https://doi.org/10.5281/zenodo.8369564). Flight trajectory and aircraft fuel consumption data are commercially sensitive, and the flight-waypoint and flight-summary outputs from GAIA can be made available for research purposes only upon reasonable request (m.stettler@imperial.ac.uk). This document used elements of Base of Aircraft Data (BADA) Family 4 Release 4.2 which has been made available by EUROCONTROL to Imperial College London.

EUROCONTROL has all relevant rights to BADA. ©2019 The European Organisation for the Safety of Air Navigation (EUROCONTROL). EUROCONTROL shall not be liable for any direct, indirect, incidental, or consequential damages arising out of or in connection with this document, including the use of BADA. This document contains Copernicus Climate Change Service information 2022. Neither the European Commission nor ECMWF is responsible for any use of the Copernicus information.

**Competing interests**

There are no conflicts of interest and all funding sources have been acknowledged. All figures are our own. None of the authors has any competing interests.

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
