# Peer review of "A high-resolution Global Aviation emissions Inventory based on ADS-B (GAIA) for 2019 – 2021"

_EGUsphere, 2023_

## Referee Comment (RC1)

**Referee comment for the review of the article "A high-resolution Global Aviation emissions Inventory based on ADS-B (GAIA) for 2019–2021"**

**General comments**

1. Does the paper address relevant scientific questions within the scope of ACP?

Aviation emission inventory datasets serve as input data for climate models to complete the background concentration of gaseous and particle emissions and atmospheric composition, are necessary to quantify the climate impact of air traffic or to assess the air quality, so the paper's content is relevant for climate research and environmental research community and fits to the scope of ACP.

2. Does the paper present novel concepts, ideas, tools, or data?

Due to the application of real waypoint data from ADS-B on a global scale an emission inventory with a more realistic routing has been created that enhances spatial and temporal accuracy and thus also regard effects from Air Traffic Management. As there were analysed the three years 2019, 2020 and 2021, the impact of the COViD-19 pandemic on air traffic volume and emissions could be quantified on a global and regional scale and compared with the pre-pandemic situation in 2019. The calculation scheme and the input data to derive the nvPM emissions, that was applied in this study, is still novel, as the input data and the methodology have been published during last 2 years.

3. Are substantial conclusions reached?

Beyond the annual aviation emission totals for the global air traffic for 2019, 2020 and 2021, the effects of the COViD-19 restrictions in 2020 and the subsequent recovery on air traffic volume and emissions have been quantified for the whole world and the dominant regional domains Europe, East Asia and USA. A subsequent evaluation of the methodology and results with a related study support their reliability. The generated results of the worldwide emission quantities, their regional and vertical distribution, annual changes during COViD-19, and the analyses of lateral inefficiencies of the flight routes provide essential information for the scientific community with focus on climate research and air traffic environmental research, but also the aeronautic industry. The final case study gives an impression of the differences of inefficiencies and emissions depending on eastbound and westbound direction affected by Jetstream.

4. Are the scientific methods and assumptions valid and clearly outlined?

The applied methodology of simulating aircraft performance and fuel flow based on BADA models and Total-Energy-Model is state-of-the-art, the majority of defined region domains was already successfully applied in other studies and the made assumptions with respect to missing emission indices and the quality check of ADS-B data and extrapolation of waypoints due to data gaps seem plausible to the referee.

5. Are the results sufficient to support the interpretations and conclusions?

Annual air traffic routing data of three years with more than 100 million flights is a robust sample to assess the changes in traffic volume, emission quantities, lateral routing and vertical distribution from year to year on a global and regional scale and separated by short-, medium- and long-haul

flights. The detailed investigation of the single route London ⟷ Singapore consists of a sample of 8705 flights over 3 years (in average ~4 per day and direction) that should be sufficient for robust statistical analysis.

6. Is the description of experiments and calculations sufficiently complete and precise to allow their reproduction by fellow scientists (traceability of results)?

As the referee also works on simulation of global 4D aircraft emission inventory datasets he is familiar with all mentioned input datasets and so presented procedure e.g. how to define the regions and how to extrapolate emissions to cruise phase is well known to the referee. The detailed presented methodology in the SI document how to deal with data gaps in ADS-B data is comprehensible and should cover the majority of possible data errors.

7. Do the authors give proper credit to related work and clearly indicate their own new/original contribution?

The article starts with a long section where recently published studies with similar scope and comparable datasets were introduced with respect to their approach and the respective similarities and differences outlined. The novel aspects of this study have been emphasized. In the results section, there is subsection 3.3 dedicated to compare the results quantitively with a study with a similar scope, that has just recently been published and addresses the same time horizon and also used real routes data.

8. Does the title clearly reflect the contents of the paper?

The title contains enough relevant keywords to allocate the content of the paper and contain aviation emissions, ADS-B data and the covered time horizon.

9. Does the abstract provide a concise and complete summary?

The abstract is a short and concise summary that lists most eye-catching results and conclusions in the same chronology as presented in the paper.

10. Is the overall presentation well structured and clear?

The paper is structured well and a clear storyline is noticeable.

11. Is the language fluent and precise?

The text is clearly verbalized and the key statements could be understood immediately.

12. Are mathematical formulae, symbols, abbreviations, and units correctly defined and used?

All abbreviations in the article were introduced when mentioned first. The introduced formulae are presented in the SI document and all mentioned variables were introduced and named. Only the parameter air traffic density (Figure 2) is recommended to be introduced in the paper as for the referee the unit km-1 h-1 is not completely clear.

13. Should any parts of the paper (text, formulae, figures, tables) be clarified, reduced, combined, or eliminated?

The presentation of the methodology, the results and the discussion are already well compressed to the essential methodology and results within the main article, a lot of detailed description is shifted to the SI.

14. Are the number and quality of references appropriate?

The number and the quality of the cited references is appropriate for the extent and scope of the study and the used input data. Most cited literature is peer-reviewed scientific journal articles; a few technical descriptions are documentations and manuals of datasets or applied models (e.g. Cirium, IEA, ICAO, EUROCONTROL, ECMWF) but are also essential and familiar in the scientific community with regard to aviation emission inventory modelling.

15. Is the amount and quality of supplementary material appropriate?

The attached material helps the referee a lot to understand the methodology in detail with regard to the quality check and processing of the ADS-B data, to get an overview of the quality of the available routing data and the allocation of the load factor and how nvPM emissions were derived. It also gives readers the opportunity to access the detailed numbers of all analysed traffic volume and emissions data also for the years 2020 and 2021, where in the main article mainly only changes relative to 2019 are presented.

**Specific comments**

1. ADS-B data provide information on the ground distance. To simulate the aircraft performance correctly information on Mach-Number or the True Air Speed (TAS) of the aircraft is necessary. Is the ERA-5 atmosphere data also used to convert ground speed to true air speed by modifying ground distance with heading information and 4D real wind speed data? Or was the assumption making that ground speed = air speed and wind effects on aircraft performance neglected? As differences of EIs and inefficiency distribution with regard to head and tail wind effects are presented in detail (Figure 7), the referee recommends to explain how the TAS/Mach number of the total energy model simulation will match the waypoint profile segments based on ground distance.
2. The air traffic density in this article is announced with the dimension [km$^{-1}$ h$^{-1}$]. The referee recommends to introduce how the air traffic density is defined in this study, as it would be generally possible to define the air traffic density as e.g. the number of aircraft movements or passenger by area or volume unit and time unit.
3. The mass of NOx emissions is a mixture of several nitroxide gases and quantified as commonly nitrogen monoxide or nitrogen dioxide mass equivalent. The referee recommends to mention within the article in which way the mass of NOx emissions should be interpreted with regard to molar mass.
4. Figure 2 compares the annual mean air traffic density of the year 2019 with the monthly mean air traffic density of 2020, the global lockdown month including depicting the difference. The referee wonders if a comparison between April 2020 and April 2019 as the same period of the last year would be better in order to isolate the Lockdown effects and to improve comparability as other seasonal and interannual effects of air traffic volume would be excluded.
5. Does the dimensionless density, shown in the colorbar of Figure 5 results from a normalization with the annual total NOx emission from GAIA and Quadros et al. (2022), respectively, to enable comparability of the latitude-altitude pattern? Summing up density values of all shown grid cells would be 1?

6.  Figure 7 e shows the distribution of fuel consumption per passenger-km. The referee wonders, where the information of passenger kilometre came from? The methodology of allocation of seat load factor was clearly described, but where does the number of seats for each aircraft come from? The absolute seat capacity would be required to derive the number of passengers for each mission, as described in SI document line 319. Or was alternatively an average seat number for the route London – Singapore assumed and obtained from Cirium database seat numbers? It would be helpful for understanding to mention this at least in the supporting information.
7.  Supporting Information document, lines 203 – 205: Summing up the relative ratios of engine types, 75%, 9% and 15% will be 99% in total. Did there happen an inaccuracy due to rounding?

**Final assessment: Minor revision**

---

## Author Comment (AC1)

**Response to Reviewer Comments**

We thank the reviewers for their comments, which helped to improve the quality and clarity of the manuscript. Below, the reviewer's comments are repeated in the *italic text*. Our response follows in normal letters. Blue text is used to cite from the revised manuscript. When page and line numbers are specified, they refer to the clean version of the revised manuscript.

**REFEREE 1 (RC1)**

**Specific comments**

1. *ADS-B data provide information on the ground distance. To simulate the aircraft performance correctly information on Mach-Number or the True Air Speed (TAS) of the aircraft is necessary. Is the ERA-5 atmosphere data also used to convert ground speed to true airspeed by modifying ground distance with heading information and 4D real wind speed data? Or was the assumption making that ground speed = air speed and wind effects on aircraft performance neglected? As differences of EIs and inefficiency distribution with regard to head and tail wind effects are presented in detail (Figure 7), the referee recommends to explain how the TAS/Mach number of the total energy model simulation will match the waypoint profile segments based on ground distance.*

   - The reviewer is correct that the 3D position of the aircraft provided by ADS-B telemetry is used to calculate the ground distance, and the ground speed (GS) is estimated by dividing the ground distance with the time elapsed between waypoints. The GS at each waypoint is subsequently converted to true airspeed (TAS) by using the magnitude of eastward and northward wind provided by the ERA5 reanalysis. The estimated TAS is then provided as inputs to the aircraft performance and emission models. Therefore, the estimated aircraft performance parameters and emission indices account for the head and tail wind effects.

   - Thank you for highlighting this missing information. A detailed description in deriving the TAS and Mach number is now included in the main text and Supporting Information (SI):

     o [Main text: Lines 152 – 154] "The ambient temperature and horizontal wind components are required to calculate the true airspeed and Mach number at each waypoint, **c.f. Eq. (S1) and (S2) in the SI §S1.1,** and we obtain  **the local meteorology** by performing a quadrilinear interpolation against historical weather data from the ERA5 HRES reanalysis."

     o [SI: Lines 70 – 82] "**We also use the aircraft GPS position (longitude and latitude) provided by the ADS-B telemetry to calculate the segment length between waypoints. The ground speed (GS) is estimated by dividing the segment length by the time elapsed between waypoints, and a Savitzky-Golay filter is used to reduce the noise in the derived GS (Savitzky and Golay, 1964). The smoothed GS is subsequently converted to true airspeed (TAS) using the historical wind fields provided by the European Centre for Medium-Range Weather Forecast (ECMWF) ERA5 high-resolution realisation (HRES) reanalysis (ECMWF, 2021; Hersbach et al., 2020),**

$$\mathbf{TAS} = \sqrt{(\mathbf{GS}\cos(\alpha) - U_{\mathbf{ERA5}})^2 + (\mathbf{GS}\sin(\alpha) - V_{\mathbf{ERA5}})^2}, \qquad \mathbf{(S1)}$$

     **where $U_{\mathbf{ERA5}}$ and $V_{\mathbf{ERA5}}$ are the eastward and northward winds at each waypoint that is estimated by performing a quadrilinear interpolation against the 4D wind fields provided by the ERA5 HRES, and $\alpha$ is the angle between the flight segment**

**and the longitudinal axis. The Mach number (Ma) is then computed for each waypoint,**

$$\mathrm{Ma} = \frac{\mathrm{TAS}}{\sqrt{\kappa RT}}, \tag{S2}$$

**where $\kappa$ (1.4) is the adiabatic index of air, $R$ (287.05 m$^2$ K$^{-1}$ s$^{-2}$) is the gas constant of dry air, and $T$ is the ambient temperature (in units of K) that is provided by the ERA5 HRES.**"

- o [SI: Lines 143 – 145] "On this basis, we perform a great-circle interpolation between the recorded waypoints to produce comparable segment lengths with d$t$ ranging between 40 and 60 s **and recompute the TAS and Mach number at each waypoint**."

2. *The air traffic density in this article is announced with the dimension [km-1 h-1]. The referee recommends to introduce how the air traffic density is defined in this study, as it would be generally possible to define the air traffic density as e.g. the number of aircraft movements or passenger by area or volume unit and time unit.*

- Thank you for highlighting this ambiguity. The air traffic density metric used in this study was previously defined in previous studies (Graf et al., 2012; Teoh et al., 2020) and is calculated by dividing the hourly mean flight distance flown by the regional surface area.

- We have made the following changes in the main text and Supporting Information to address this point:

  - o [Main text: Lines 249 – 250]: "Fig. 2a shows the **2019** global air traffic density **, defined as the total flight distance flown divided by the regional surface area and time**."

  - o [Footnote under Table 4 (main text) and Tables S9 and S10 (Supporting Information)]: "**The air traffic density (ATD) is defined as the total flight distance flown in the region divided by its surface area and time, $\mathrm{ATD}\,[\mathrm{km}^{-1}\,\mathrm{h}^{-1}] = \frac{\sum \text{Annual flight distance flown [km]}}{\text{Surface area [km}^2]\,\times(365\times24\,[\mathrm{h}])}$**."

3. *The mass of NOx emissions is a mixture of several nitroxide gases and quantified as commonly nitrogen monoxide or nitrogen dioxide mass equivalent. The referee recommends to mention within the article in which way the mass of NOx emissions should be interpreted with regard to molar mass.*

- Thank you for this question. According to the ICAO Annex 16 Vol II document (ICAO, 2017), the engine specific $NO_X$ emission indices (EI) are reported as an $NO_2$ mass equivalent which is now clarified in the manuscript. In addition, we also now provide a recommendation for future studies to break down the reported $NO_X$ emissions into individual species of nitric oxide (NO), nitrogen dioxide ($NO_2$), and nitric acid (HONO).

- The following changes have been made to the main text to address these points:

  - o [Main text: Lines 34 – 37]: "$NO_X$**,**  **which includes both nitric oxide (NO) and nitrogen dioxide ($NO_2$) gases, emitted in the stratosphere facilitates the production of: (i) ozone, which causes a warming effect; and (ii) hydroxyl radicals, which partly offsets this warming effect through the destruction of methane** (Fuglestvedt et al., 1999; Myhre et al., 2011)."

  - o [Main text: Lines 194 – 201]: "**We also highlight that the engine-specific $NO_X$ EI in the ICAO EDB is reported as an $NO_2$ mass equivalent (ICAO, 2017). For future**

**studies that require cruise $NO_X$ emissions to be broken down into individual species, references can be made to previous in-situ measurements which assumes the engine exit $NO_2/NO_X$ and $NO/NO_X$ molar mixing ratio to have a global mean of 0.07 and 0.93 respectively (Schulte et al., 1997), and the nitrous acid (HONO) EI to be 0.31 g per kg-$NO_2$ (Jurkat et al., 2011). For the landing and take-off cycle (LTO), existing studies have estimated that the $NO_2/NO_X$ molar mixing ratio varies significantly based on engine type and thrust settings, and ranges between: (i) 0.05–0.10 during climb and take-off; (ii) 0.12–0.20 during the descent phase; and (iii) 0.75–0.98 during the taxi phase (Timko et al., 2010; Wood et al., 2008; Wey et al., 2006; Stettler et al., 2011)."**

o [Main text: Table 3]:

| Annual statistics | 2019 | 2020 | 2021 | % Change | |
|---|---|---|---|---|---|
| | | | | 2020 vs. 2019 | 2021 vs. 2019 |
| Total number of flights | 40,221,182 | 27,911,214 | 35,576,376 | -31% | -12% |
| - Jet | 33,224,736 | 20,302,177 | 24,458,494 | -39% | -26% |
| - Turboprop | 3,231,103 | 2,719,339 | 3,754,998 | -16% | 16% |
| - Piston | 3,765,343 | 4,889,698 | 7,362,884 | 30% | 96% |
| Distance travelled ($\times 10^9$ km) | 60.94 | 34.50 | 41.90 | -43% | -31% |
| - Jet | 59.00 | 32.59 | 39.16 | -45% | -34% |
| - Turboprop | 1.34 | 1.13 | 1.56 | -15% | 17% |
| - Piston | 0.61 | 0.78 | 1.18 | 29% | 94% |
| Mean passenger load factor (%)[a] | 83% | 59% | 67% | -29% | -19% |
| Mean aircraft mass (kg) | 64079 | 49593 | 46533 | -23% | -6.2% |
| Fuel burn (Tg) | 283 | 146 | 166 | -48% | -41% |
| Fuel burn per distance (kg $km^{-1}$) | 4.636 | 4.240 | 3.958 | -8.5% | -15% |
| $CO_2$ (Tg) | 893 | 462 | 524 | -48% | -41% |
| $H_2O$ (Tg) | 348 | 180 | 204 | -48% | -41% |
| OC (Gg) | 5.65 | 2.93 | 3.32 | -48% | -41% |
| $SO_2$ (Gg) | 339 | 176 | 199 | -48% | -41% |
| $S^{VI}$ (Gg) | 6.92 | 3.58 | 4.06 | -48% | -41% |
| $NO_X$ (**as $NO_2$,** Tg) | 4.49 | 2.26 | 2.55 | -50% | -43% |
| CO (Gg) | 400 | 227 | 272 | -43% | -32% |
| HC (Gg) | 33.9 | 20.9 | 25.0 | -38% | -26% |
| nvPM mass (Gg) | 21.4 | 9.93 | 11.0 | -54% | -49% |
| nvPM number ($\times 10^{26}$) | 2.83 | 1.46 | 1.66 | -48% | -41% |
| Mean EI $NO_X$ (g $kg^{-1}$) | 15.9 | 15.4 | 15.4 | -2.8% | -3.2% |
| Mean EI CO (g $kg^{-1}$) | 1.42 | 1.55 | 1.64 | 9.6% | 16% |
| Mean EI HC (g $kg^{-1}$) | 0.120 | 0.143 | 0.151 | 19% | 26% |
| Mean nvPM $EI_m$ (g $kg^{-1}$) | 0.076 | 0.068 | 0.066 | -10.4% | -12% |
| Mean nvPM $EI_n$ ($\times 10^{15}$ $kg^{-1}$) | 1.002 | 0.998 | 1.001 | -0.4% | -0.1% |

[a]: The passenger load factor for each flight was derived using the global and regional data published by ICAO and IATA (refer to Section 2.2 and the SI §S3).

4. *Figure 2 compares the annual mean air traffic density of the year 2019 with the monthly mean air traffic density of 2020, the global lockdown month including depicting the difference. The referee wonders if a comparison between April 2020 and April 2019 as the same period of the last year would be better in order to isolate the Lockdown effects and to improve comparability as other seasonal and interannual effects of air traffic volume would be excluded.*

- Thank you for this suggestion. Figure 2 shows the: (a) annual air traffic density for 2019; (b) the monthly air traffic density in April-2020 where global air traffic is at a minimum; and (c) a comparison between the change in global air traffic density between the full year of 2019 and 2020.

- The main rationale for plotting the 2019 global air traffic density in (a) and the April-2020 global air traffic density in (b) is because these most clearly illustrate the impact of COVID-19 on global aviation activity. We note that the air traffic density metric, now defined in Point (2), is normalised against time. Therefore, the air traffic density metric is independent of the timeframe that is selected for comparison (i.e., annual vs. monthly).

- The main rationale for comparing the annual change in air traffic density between the full year 2019 and 2020 in (c) is to highlight the prolonged impact of COVID-19 in different countries. For example, a comparison of the full year statistics clearly show that China is the only country that experienced growth in air traffic activity. We experimented with the referee's suggestion of comparing the change in monthly air traffic density between April-2019 and April-2020. However, unlike the annual comparison (in the original figure), it does not show the increase in air traffic density over China during this month (figure below). Therefore, we have decided to keep the original Figure 2 unchanged.

[Figure]

[Figure]

- We have updated the caption of Figure 2 for clarity improvements:

    [Main text: Lines 286 – 288]: "The global **(a) annual** air traffic density in  2019;  (b) **monthly air traffic density in** April-2020, where air traffic activity was at a minimum due to the COVID-19 pandemic; and (c) the change in **annual** air traffic density between 2019 and 2020. Basemap plotted using Cartopy 0.21.1 © Natural Earth; license: public domain."

5. *Does the dimensionless density, shown in the colorbar of Figure 5 results from a normalization with the annual total NOx emission from GAIA and Quadros et al. (2022), respectively, to enable comparability of the latitude-altitude pattern? Summing up density values of all shown grid cells would be 1?*

    - Thank you for highlighting this. Yes, the reviewer is correct that the value at each grid cell is normalised with the global annual emissions provided by each dataset.

    - We have updated the caption of this figure (now Figure 6 in the main text) for clarity improvements:

    - [Main text: Lines 357 – 361]: "Figure 6: Distribution of the 2019 $NO_X$ and nvPM number emissions by latitude and altitude from GAIA with actual flight trajectories (subplots a and c) versus estimates from Quadros et al. (2022) with monthly-averaged flight trajectories (subplots b and d). **The colour bar represents the proportion of annual emissions, i.e., annual emissions at each grid cell divided by the global annual emissions, where the summation of values across all grid cells would be equal to 1.**"

6. *Figure 7e shows the distribution of fuel consumption per passenger-km. The referee wonders, where the information of passenger kilometre came from? The methodology of allocation of seat load factor was clearly described, but where does the number of seats for each aircraft come from? The absolute seat capacity would be required to derive the number of passengers for each mission, as described in SI document line 319. Or was alternatively an average seat number for the route London – Singapore assumed and obtained from Cirium database seat numbers? It would be helpful for understanding to mention this at least in the supporting information.*

   - Thank you for highlighting this. We agree that this information is important and a short description of the methodology to estimate the fuel consumption per passenger-km is now included the Supporting Information (SI):

     o [SI: Lines 506 – 515] "**Fig. 9 in the main text highlights the variability in flight trajectory, fuel consumption and emissions for eastbound and westbound flights between London Heathrow (LHR) and Singapore Changi Airport (SIN) in 2019-2021, totalling 8705 unique flights. During this time, the three main aircraft types used for this route are the Boeing 777 (40.8% of all flights), Airbus A380 (38.6%), and the Airbus A350 (20.6%); and the three main airline operators are Singapore Airlines (65.0% of all flights), British Airways (23.9%), and Qantas Airways (8.9%). For each flight, the fuel consumption per passenger-km is calculated as follows:**

     $$\text{Fuel per passenger-km} = \frac{\text{Total fuel burn}}{(\text{Aircraft seat capacity} \times \text{Passenger LF}) \times \text{Flight distance flown}}, \quad \textbf{(S8)}$$

     **where the registered seat capacity for each unique aircraft is provided by the Cirium global fleet database (Cirium, 2022), while the methodology to estimate the passenger LF is listed in the SI §S3.**"

     o [Main text: Lines 421 – 425] "Figure 9: The (a) lateral and (b – c) vertical trajectory that is flown by flights between London Heathrow Airport (LHR) and Singapore Changi Airport (SIN) between 2019 and 2021 (n = 8705), and probability density function of the (d) lateral inefficiency, (e) fuel consumption per passenger-km; (f) EI $NO_X$; and (g) nvPM $EI_n$ for westbound (SIN–LHR, in blue) and eastbound flights (LHR–SIN, in red). **The methodology to estimate the fuel consumption per passenger-km is described in the SI §S5.2.**"

7. *Supporting Information document, lines 203 – 205: Summing up the relative ratios of engine types, 75%, 9% and 15% will be 99% in total. Did there happen an inaccuracy due to rounding?*

   - Thank you for identifying this. Yes, the reviewer is correct that the discrepancy is caused by rounding error. To rectify this, we have now quoted the figures in the SI §S1.3 to one decimal place:

     o [SI: Lines 215 – 247]: "Fig. S7 presents the summary statistics for the cleaned ADS-B dataset and shows that:

       - 103.7 million flight trajectories are recorded between 2019 and 2021 (Fig. S7a),

       - 75.**2**% of all flights are carried out by jet aircraft, 9.**4**% by turboprops, and the remaining 15.**4**% by piston aircraft (Fig. S7b),

       - origin and destination airport metadata are available for 79.**1**% of all flights, and this figure increases to 90.9% when piston aircraft, mostly used in general aviation, are excluded (Fig. S7c),

       - 67.**4**% of all flights have full trajectory coverage, i.e., first waypoint starting from the origin airport and ending at the destination airport, and this figure increases to 7**7.6**% when piston aircraft are excluded (Fig. S7d),

       - 5.0% of all flights are rejected from the ADS-B dataset (Fig. S7e), and

- at the waypoint level, 99.**5**% of the recorded ADS-B signals are from terrestrial receivers and the remaining **0.5**1% are provided by satellite receivers (Fig. S7f).

The 5% of all flights that are rejected from the ADS-B dataset are caused by identified errors in their respective flight trajectories, for example,

- trajectories that contain less than three waypoints (57**6.6**% of all rejected flights),
- trajectories with very long extrapolated segment lengths, i.e., > 90% of the distance between the origin-destination airport (21**0.6**% of all rejected flights),
- flights with unrealistic flight time (13.**3**% of all rejected flights), and
- flight segments with unrealistic ground speed (**9.**5% of all rejected flights)."

**REFEREE 2 (RC2)**

*This manuscript presents a new aviation emissions data based on individual flight level data and models of engine performance. Most of the technical details are in the supplement, and summary analysis is presented in the main manuscript.*

*The manuscript is a good description and analysis of an important new dataset for aviation emissions covering the period 2019-2021, pre, beginning and mid pandemic. The analysis in the manuscript is well presented. It should be publishable with minor revisions. I think some of the analysis could be presented a little better with some more figures (in addition to tables) as noted below. The supplement seems correct and comprehensive. The data (low time resolution) has been posted to an available archive.*

**Specific comments:**

8. *Page 2, L31: where does the 1034 Tg number come from (reference?)*

   - The source of the 1034 Tg of $CO_2$ emissions, amounting to 2.4% of the anthropogenic greenhouse gas, was quoted from Lee et al. (2021), which was already cited at the end of the sentence.

9. *Page 5, L140: How does the -8% lower comparison at major airports mesh with +15% to +20% globally?*

   - Thank you for the comment. In the main text, we compared the ADS-B dataset against three different sources:
     - The total number of flights in the ADS-B dataset differs by -4.7% (2019), +14% (2020), and +17% (2021), respectively, relative to the statistics from ICAO and IATA,
     - The annual flight distance flown in the ADS-B dataset are 8% (2019), 23% (2020), and 24% (2021) larger than the estimates from Airlines for America, and
     - The ADS-B dataset could underestimate the 2019-2021 air traffic movements in London Heathrow Airport (-1.3%), New York John F. Kennedy Airport (-8.1%), and Singapore Changi Airport (-1.3%).

   - For comparison (i) and (ii), the increasing global coverage area of ADS-B receiver networks over time likely lead to more flights being captured in the ADS-B dataset from 2019 to 2021. In general, statistics provided by ICAO, IATA and Airlines for America should also be lower than the ADS-B dataset because they only account for the air traffic from scheduled flights, while the ADS-B dataset captures unscheduled flights such as charter flights and private aviation.

- For comparison (iii), air traffic movements provided by the ADS-B dataset is 1–8% lower than the official airport statistics, and this can most likely be attributed to our data cleaning algorithm which rejected flights with erroneous trajectories that cannot be verified.

- We also revisited the official airport statistics (Airport Traffic Statistics, 2022) and identified a minor error in our earlier data compilation for New York John F. Kennedy (JFK) airport. The total 2019-2021 air traffic movements in JFK have been revised down from 958,420 (+8.1% relative to GAIA) to 946,390 (+7.0% relative to GAIA).

- We have made the following changes in the main text for clarity improvements:

  o [Main text: Lines 127 – 140]: "As these the statistics **from ICAO, IATA and Airlines for America** only captures air traffic activity from scheduled flights, we exclude general aviation activity in these comparisons by omitting flights that are flown by piston aircraft. The total number of flights in the ADS-B dataset differs by -4.7% in (2019), +14% in (2020), and +17% in (2021), respectively, relative to the statistics from ICAO and IATA (Table S1); while the annual flight distance flown **in the ADS-B dataset** are 8% (2019), 23% (2020), and 24% (2021) larger than the estimates from Airlines for America (Table S2). These discrepancies are likely due to: (i) an increasing **global** coverage area of ADS-B receiver networks over time **enabling more flights to be captured in the ADS-B dataset** (Fig. S1); (ii) an increase in the proportion of non-scheduled flights, i.e., charter flights, cargo services and private aviation, from 4.1% in 2019 to 7.5% in 2020 (Sobieralski and Mumbower, 2022; ICAO, 2021); and (iii) a higher occurrence of rejected flights (i.e., trajectories with less than three waypoints, unrealistic segment lengths, flight times and/or ground speeds) in 2019 (~6.6%) relative to 2020 (~3.3%) and 2021 (~4.5%) (Fig. S7e). A comparison with **data** traffic statistics from three major airports (London Heathrow, New York John F. Kennedy, and Singapore Changi Airport) suggest that the 2019–2021 air traffic movements in the ADS-B dataset are 1-8% lower than the official statistics **is 1.3%, 7.0%, and 1.3% lower than the official statistics from London Heathrow, New York John F. Kennedy, and Singapore Changi airport, respectively** (Fig. S8)**, and this discrepancy can most likely be attributed to our data cleaning algorithm which rejected flights with erroneous trajectories that cannot be verified (SI §S1.2).**"

  o [SI: Lines 280 – 285]: "Fig. S8 shows that the total number of aircraft movements derived from the processed ADS-B dataset can be between 1–7% lower when compared with published statistics from the three airports (-1.3% for EGLL, -**7.0**% for KJFK and -1.3% for WSSS between 2019 and 2021). For the comparison with WSSS, we note that the published data does not include air traffic movements from freight operations and private aviation, and therefore, the **monthly** number of flights in the ADS-B dataset can be higher than the published statistics."

10. *Page 5, L150: can you say what the default is and what the modifications are briefly?*

- Thank you for the comment. We have made the following changes in the main text for clarity improvements:

  o [Main text: Lines 149 – 151]: "For aircraft not covered by the fleet database, we assign the default aircraft-engine combination from **provided by** BADA **3** with minor modifications **applied to the Airbus A320, Boeing B787 to select the engine option with the highest market share** (Table S4)."

11. *Page 11, L255: see note on table 4. I suggest making a histogram that makes clear the differences between consumption and distance (and emissions) by region…*

- Please see our response to Point (12) below.

12. *Page 13, L290: suggest that the percent values of distance, fuel and emissions by region be made into a histogram (grouped by region) to make deviations from say % distance travelled evident.*

- Thank you for this suggestion. We have now included a bar chart in the main text to summarise the distribution of the annual flight distance flown, fuel consumption and emissions for each region:

    o [Main text: Lines 309 – 311]

[Figure]

**Figure 4: Percentage breakdown of the 2019 global annual flight distance flown, fuel consumption, NO$_X$, CO, HC, nvPM mass and number emissions by region.**

13. *Page 11, L285: maybe figure 2c could be in percent?*

- Thank you for this suggestion. We have considered plotting Figure 2c (i.e., the change in air traffic density between 2019 and 2020) in percentage terms but have decided against it. This is because of the significant spatiotemporal variation in air traffic activity, where: (i) it is not possible to calculate the percentage change in grid cells that previously had no air traffic activity in 2019 because the denominator would be zero; and (ii) there can be grid cells with a very large percentage change (> 1000%) if their air traffic activity in 2019 (denominator) is small.

14. *Page 11, L252: for fleet composition, higher fuel use rate per km is due to more narrow body aircraft?*

- Thank you for this comment. We have addressed this together with Point (15) below.

15. *Page 15, L308: I though short haul were less efficient (kg/km) per unit distance? (E.g. discussing China above) Or is that not the case? Please clarify.*

- Thank you for the questions. The comparison in the fuel efficiency between short- and long-haul flights is influenced by the selected metric, i.e., fuel consumption per passenger-km vs. the fuel consumption per flight distance flown. It is likely that long-haul flights are more efficient if the fuel consumption per passenger-km was chosen as a benchmark. However, our reported metric was the fuel consumption per flight distance flown, which is not normalised by the number of passengers.

- Short-haul flights have a lower fuel consumption per distance flown relative to long-haul flights because they are predominantly flown by narrow-body aircraft that have a lower mean aircraft mass relative to long-haul flights that are generally flown by wide-body aircraft (Tables 5, S11 and S12).

- We have made the following changes to the main text for clarity improvements:

  o [Main text: Lines 260 – 267]: "The proportion of fuel consumption in the US (19%) is lower than its share of aviation activity (27% of the global annual flight distance flown), **and** the mean  fuel consumption **per distance flown** (3.29 kg km$^{-1}$) **is** 29% lower than the global average (4.64 kg km$^{-1}$). In contrast, the North Atlantic and North Pacific flight corridors have a higher share of fuel consumption (13%) than their distance flown (8.8%), **and** the **mean** distance-specific fuel consumption (6.61 kg km$^{-1}$) **is** 43% higher than the global average (Table 3). The discrepancies in distance-specific fuel consumption are due to a higher proportion of short-haul domestic flights in the US predominantly served by smaller narrow-body aircraft, while larger **and heavier** wide-body aircraft are  **generally** used for long-haul transoceanic flights."

  o [Main text: Lines 317 – 325]: "Globally, reductions in the annual fuel consumption and $CO_2$ emissions (-48% in 2020 and -41% in 2021 relative to 2019 levels) are greater than the change in flight distance travelled (-43% and -31% respectively), and the mean  fuel consumption **per flight distance flown** in 2020 (4.24 kg km$^{-1}$) and 2021 (3.96 kg km$^{-1}$) were 9% and 15% lower than in 2019 (4.64 kg km$^{-1}$), respectively (Table 3).  **The** lower **distance-specific** fuel consumption **is most likely caused by the lower global annual mean aircraft mass (-23% in 2020 and -6.2% in 2021 relative to 2019, Table 3)** , **, which in turn, can be attributed to the**: (i)  lower **annual** mean passenger load factor **(59% in 2020 and 67% in 2021 vs. 83% in 2019)** ; (ii) a **reduction** in **long**-haul flights (> **6** h) in 2020 and 2021 that are predominantly flown by **wide**-body aircraft (**8**% of all flights) relative to 2019 (**5**%)**, where their mean aircraft mass is around 2–4 times larger than short- and medium-haul flights (< 6 h) that are generally flown by narrow-body aircraft (Tables 5, S11 and S12)."**

16. *Page 20, L409: You have Fuel consumption per passenger-km: is that the same as gross fuel consumption? Presumably the rerouting is to save fuel, so the variance in fuel used should be smaller than distance? Would be interesting maybe to state total fuel per flight on this route and how much it varies (more or less than distance).*

- Thank you for this suggestion. To answer this question, we have performed additional post-analysis of the GAIA outputs by grouping all flights in 2019 by their origin-destination (OD) airport pairs. For each OD airport pairs, we then compare the: (i) actual flight distance flown ($d_{GAIA}$) versus the great circle distance between the OD airport ($d_{OD}$); and (ii) simulated fuel consumption from GAIA ($f_{GAIA}$) versus an estimate of the fuel consumption from the great circle trajectory that is derived from the European Environment Agency (EEA) emissions calculator ($f_{EEA,OD}$). We note that the EEA calculates fuel consumption as a function of $d_{OD}$ and aircraft type. An evaluation across all OD airport pairs globally show that the variability of $f_{GAIA}/f_{EEA,OD}$ (1.14 [0.997, 1.35], i.e., mean of 1.14 and a 5$^{th}$ and 95$^{th}$ percentile of 0.997 and 1.35 respectively) is greater than $d_{GAIA}/d_{OD}$ (1.06 [1.01, 1.16]), and this can most likely be attributed to the: (i) use of different aircraft types to complete the same mission; and the day-to-day variability in (ii) passenger load factor; and (iii) ambient wind fields.

- For the 8,705 flights between London and Singapore, we also evaluated the difference in coefficient of variation (CV) between the flight distance flown versus the fuel consumption per passenger-km. Our results show that the CV of the fuel consumption per passenger-km (0.171) is around 8 times larger than the CV of the flight distance flown (0.021), and this is most likely

due to: (i) the use of different aircraft-engine types; (ii) variabilities in aircraft seating capacity between airlines; as well as the day-to-day variability in the (iii) passenger load factor; and (iv) ambient wind conditions.

- We have now included these analyses in the main text and Supporting Information:

  - [Main text: Lines 403 – 414]: "**We group each flight in 2019 by their origin-destination (OD) airport pairs to evaluate the differences in their mean: (i) historical flight distance flown ($d_{GAIA}$) versus $d_{OD}$; and (ii) simulated fuel consumption from GAIA ($f_{GAIA}$) versus the fuel consumption from the great circle trajectory that is estimated from the European Environment Agency emissions calculator ($f_{EEA,OD}$), where $f_{EEA,OD}$ at climb, cruise, and descent (CCD) for each flight is estimated as a function of $d_{OD}$ and aircraft type (European Environment Agency, 2019). These OD statistics have been made open source and are available, as described in the Data Availability statement. In general,** there is an inverse relationship between the lateral inefficiency and **, where** the mean lateral inefficiency is 5.1 [0.7, 10.8] % (5[th] and 95[th] percentile) for flights with $d_{OD}$ < 1000 km, 4.4 [0.9, 10.2] % when $d_{OD}$ is between 1000 and 2000 km, and 2.9 [0.8, 8.6] % for $d_{OD}$ > 2000 km **(Fig. 7b and Fig. 8a). Our analysis among OD airport pairs also suggests that: (i) the variability of $f_{GAIA}/f_{EEA,OD}$ (1.14 [0.997, 1.35]) is greater than $d_{GAIA}/d_{OD}$ (1.06 [1.01, 1.16]) (Fig. S15); (ii) $d_{GAIA}/d_{OD}$ is relatively symmetrical between OD airport pairs irrespective of direction travelled (Fig. 8a); and (iii) there is a directional bias in $f_{GAIA}/f_{EEA,OD}$ among OD airport pairs, e.g., westbound transatlantic flights to consume more fuel than eastbound transatlantic flights (Fig. 8b), because $f_{GAIA}$ captures the effects of ambient wind patterns.**"

[Figure]

**Figure 1: The mean ratio of: (a) the actual flight distance flown from GAIA over the great circle distance ($d_{GAIA}/d_{OD}$); and (b) the simulated fuel consumption from GAIA over the estimated fuel consumption from the great circle trajectory ($f_{GAIA}/f_{EEA,OD}$) across all flights that traversed between major European and North American airport pairs in 2019.**

  - [SI: Lines 487 – 505] "**Each flight in 2019 is also grouped by their origin-destination (OD) airport pairs and corresponding countries to evaluate the difference in their mean: (i) historical flight distance flown ($d_{GAIA}$) versus the great circle trajectory between the origin-destination airport pairs ($d_{OD}$); and (ii) simulated fuel consumption from the actual flight trajectory in GAIA ($f_{GAIA}$) versus the estimated fuel consumption at climb, cruise, and descent (CCD) from the great circle trajectory ($f_{EEA,OD}$) which is derived from an emissions calculator developed by the European Environment Agency using inputs of $d_{OD}$ and aircraft type (European Environment Agency, 2019). These statistics, which vary significantly between OD airport pairs**

**(Fig. 8 in the main text), have been made publicly available as described in the Data Availability statement (main text). Across all OD airport pairs, we estimate a mean $d_{GAIA}/d_{OD}$ of 1.06 [1.01, 1.16] (5th and 95th percentile) and a mean $f_{GAIA}/f_{EEA,OD}$ of 1.14 [0.997, 1.35] (Fig. S15). We also note that the variability of $f_{GAIA}/f_{EEA,OD}$ is greater than $d_{GAIA}/d_{OD}$, and this can most likely be attributed to the: (i) use of different aircraft types (i.e., narrow- and wide-body aircraft) to complete the same mission; and the day-to-day variability in (ii) passenger load factor (LF); and (iii) ambient wind fields (i.e., headwind and tailwind)."**

[Figure]

**Figure S2: Kernel density estimate between the mean ratios of $f_{GAIA}/f_{EEA,OD}$ and $d_{GAIA}/d_{OD}$ for each origin-destination airport pairs globally in 2019 (n = 36,626).**

- [SI: Lines 515 – 519] **"We note that the coefficient of variation (CV), i.e., the ratio of the standard deviation to the mean, of the fuel consumption per passenger-km (0.171) is around 8 times larger than the CV of the flight distance flown (0.021), which most likely arises from: (i) the use of different aircraft-engine types; as well as the day-to-day variability in (ii) the passenger LF; and (iii) ambient wind conditions."**

**REFERENCES**

[revised manuscript text omitted]